# Total parasite biomass but not peripheral parasitaemia is associated with endothelial and haematological perturbations in *Plasmodium vivax* patients

**João L Silva-Filho[1,2]\*[†], João CK Dos-Santos[1,3][†], Carla Judice[1], Dario Beraldi[2], Kannan Venugopal[2], Diogenes Lima[4][‡], Helder I Nakaya[4,5], Erich V De Paula[6], Stefanie CP Lopes[6,7,8], Marcus VG Lacerda[7,8], Matthias Marti[2]\*, Fabio TM Costa[1]\***

[1]Laboratory of Tropical Diseases – Prof. Luiz Jacintho da Silva, Department of Genetics, Evolution, Microbiology and Immunology, Institute of Biology, University of Campinas, Campinas, Brazil; [2]Wellcome Centre for Integrative Parasitology, Institute of Infection, Immunity & Inflammation, University of Glasgow, Glasgow, United Kingdom; [3]Post-Graduation in Medical Pathophysiology, School of Medical Sciences, University of Campinas, Campinas, Brazil; [4]School of Pharmaceutical Sciences, University of São Paulo, São Paulo, Brazil; [5]Hospital Israelita Albert Einstein, São Paulo, Brazil; [6]Department of Clinical Pathology, School of Medical Sciences, University of Campinas, Campinas, Brazil; [7]Institute Leônidas & Maria Deane, Fiocruz, Manaus, Brazil; [8]Tropical Medicine Foundation Dr. Heitor Vieira Dourado, Manaus, Brazil

**\*For correspondence:**
joao.dasilvafilho@glasgow.ac.uk (JLS-F);
matthias.marti@glasgow.ac.uk (MM);
fabiotmc72@gmail.com (FTMC)

[†]These authors contributed equally to this work

[‡]Deceased

**Abstract** *Plasmodium vivax* is the major cause of human malaria in the Americas. How *P. vivax* infection can lead to poor clinical outcomes, despite low peripheral parasitaemia, remains a matter of intense debate. Estimation of total *P. vivax* biomass based on circulating markers indicates existence of a predominant parasite population outside of circulation. In this study, we investigate associations between both peripheral and total parasite biomass and host response in vivax malaria. We analysed parasite and host signatures in a cohort of uncomplicated vivax malaria patients from Manaus, Brazil, combining clinical and parasite parameters, multiplexed analysis of host responses, and ex vivo assays. Patterns of clinical features, parasite burden, and host signatures measured in plasma across the patient cohort were highly heterogenous. Further data deconvolution revealed two patient clusters, here termed Vivax[low] and Vivax[high]. These patient subgroups were defined based on differences in total parasite biomass but not peripheral parasitaemia. Overall Vivax[low] patients clustered with healthy donors and Vivax[high] patients showed more profound alterations in haematological parameters, endothelial cell (EC) activation, and glycocalyx breakdown and levels of cytokines regulating different haematopoiesis pathways compared to Vivax[low]. Vivax[high] patients presented more severe thrombocytopenia and lymphopenia, along with enrichment of neutrophils in the peripheral blood and increased neutrophil-to-lymphocyte ratio (NLCR). When patients' signatures were combined, high association of total parasite biomass with a subset of markers of EC activation, thrombocytopenia, and lymphopenia severity was observed. Finally, machine learning models defined a combination of host parameters measured in the circulation that could predict the extent of parasite infection outside of circulation. Altogether, our data show that total parasite biomass is a better predictor of perturbations in host homeostasis in *P. vivax* patients than peripheral

parasitaemia. This supports the emerging paradigm of a *P. vivax* tissue reservoir, particularly in the haematopoietic niche of bone marrow and spleen.

## Introduction

Malaria remains a heavy burden across endemic regions worldwide. In 2018, *Plasmodium vivax* infection accounted for 41% of all malaria cases outside of Sub-Saharan Africa, with a total of 6.5 million cases and more than 2 billion people in 90 countries at risk (***World Malaria Reports, 2019***). There are concerns that *P. vivax* elimination will be significantly more difficult than *P. falciparum* as the current measures for malaria control are less effective for *P. vivax* than for *Plasmodium falciparum*, with the elimination of the former presenting a major challenge in areas that successfully reduced *P. falciparum* burden. This persistence is due to some unique biological features complicating treatment and elimination, including low peripheral parasitaemia and presence of dormant liver stages (hypnozoites) which relapse weeks or months after blood infection has been cleared.

*P. vivax* infection is associated with low peripheral parasitaemia (<2%) as a result of a strict host cell tropism to immature reticulocytes (***Malleret et al., 2015***; ***Mayor and Alano, 2015***) that are exceedingly rare in peripheral blood (<2%) but highly prevalent in the haematopoietic niche of bone marrow (BM) and spleen (***Klei et al., 2017***; ***Rhodes et al., 2016***). Because of limited microvascular adherence in vivo and endothelial cell (EC) binding in vitro (***Lacerda et al., 2012***; ***Valecha et al., 2009***), it was generally assumed that peripheral parasitaemia reflects the majority of *P. vivax* parasites during infection. However, discrepancy of parasite biomass based on systemic biomarkers such as *Plasmodium* lactate dehydrogenase (pLDH) compared to peripheral parasitaemia supports existence of a major *P. vivax* reservoir outside of circulation, particularly in patients with complicated outcomes (***Barber et al., 2015***). In support of this hypothesis, studies have demonstrated that late asexual blood stage *P. vivax* parasites (i.e. schizonts) are capable of cytoadhering to endothelial host receptors (***Carvalho et al., 2010***; ***De las Salas et al., 2013***) and present at reduced abundance compared to the other blood stages in the blood of *P. vivax* patients (***Obaldia et al., 2018***; ***Lopes et al., 2014***). In experimentally infected non-human primates (NHPs), a significant enrichment of sexual stages (gametocytes) and schizonts in BM sinusoids and parenchyma has been observed (***Obaldia et al., 2018***), supporting previous evidence from multiple case reports that identified *P. vivax* in BM and spleen (***Yx et al., 2009***; ***Wickramasinghe et al., 1989***; ***Wickramasinghe and Abdalla, 2000***; ***Salutari et al., 1996***; ***Baro et al., 2017***; ***Machado Siqueira et al., 2012***; ***Lacerda et al., 2008***; ***Brito et al., 2020***). A series of recent studies in acute and chronic human *P. vivax* infection have meanwhile provided direct evidence that BM and spleen represent the major reservoir of parasite biomass in *P. vivax* infection (***Baro et al., 2017***; ***Brito et al., 2020***; ***Kho et al., 2021a***; ***Kho et al., 2021b***).

*P. vivax* parasites can elicit a potent host response, including inflammation and EC activation, and cause severe and fatal manifestations at significantly lower peripheral parasitaemia than the more virulent species, *P. falciparum* (***Barber et al., 2015***; ***Yeo et al., 2010***). However, the pathogenic mechanisms underlying these alterations in host homeostasis and their relationship with *P. vivax* biomass are not fully understood (***Lacerda et al., 2011***; ***Naing and Whittaker, 2018***; ***Rodriguez-Morales et al., 2005***; ***Tangpukdee et al., 2008***).

Here we systematically investigated host responses in a cross-sectional cohort of uncomplicated *P. vivax* patients from Manaus, in the Brazilian Amazon region. Our analysis revealed an association between alterations in host homeostasis, including EC activation, damage, and haematological disturbances, such as thrombocytopenia, lymphopenia, and increased neutrophils turnover, with total parasite biomass but not peripheral parasitaemia. These findings are in line with the emerging paradigm of a clinically relevant parasite reservoir outside of circulation and merit systematic investigations of this reservoir in vivax malaria.

## Results

### Uncomplicated *P. vivax* patients present with haematological changes

We have conducted a cross-sectional study with uncomplicated *P. vivax* malaria patients seen at FMT-HVD in Manaus, Brazil. We included 79 adult patients (median age of 36 years) with confirmed *P. vivax* infection (smear and PCR positive) and 34 age- and sex-matched uninfected healthy donors

**Table 1.** Demographic, parasite, and multiplexed microbead-based immunoassay (Luminex) data obtained from the plasma of a representative subset of 31 *P. vivax* patients and 9 healthy donors (controls).

| Parameters | Healthy donors (n = 36) | Symptomatic Pv patients (n = 79) | p-Value (Pv vs. control) |
|---|---|---|---|
| | Median [IQ 25–75] | Median [IQ 25–75] | |
| Age | 32 (23–49) | 36 (28–45) | 0.06 |
| Parasitaemia (103 /mL) | - | 4.29 [1.86–6.62] | |
| Parasitaemia (%) | - | 0.76 [0.57–1.25] | |
| Parasite load (copies 18S RNA/mL) | - | 26,642 [9253-522,297] | |
| PvLDH (O.D.) | - | 0.18 [0.005–0.34] | |
| **Plasma biomarkers** | **Healthy donors (n = 9)** | **Symptomatic Pv patients (n = 31)** | **p-Value (Pv vs. control)** |
| TNF-α (pg/mL) | 17.2 [11.0–22.3] | 38.4 [30.0–69.6] | <0.0001 |
| IL-1α (pg/mL) | 11.9 [10.0–19.5] | 25.4 [19.8–33.5] | 0.0004 |
| IL-1β (pg/mL) | 12.0 [8.0–12.8] | 21.4 [14.5–27.6] | <0.0001 |
| IL-6 (pg/mL) | 3.0 [2.5–3.7] | 33.4 [7.6–133.1] | <0.0001 |
| IL-8 (pg/mL) | 2.2 [0.6–2.4] | 6.4 [2.7–19.9] | 0.0005 |
| IL-10 (pg/mL) | –* | 314 [169–562] | – |
| G-CSF (pg/mL) | 9.485 [9.485–9.485] | 101.5 [33.49–239.6] | <0.0001 |
| L-selectin (ng/mL) | 326 [287–391] | 481 [386–579] | 0.0019 |
| ICAM-1 (ng/mL) | 323 [260–464] | 634 [456–849] | 0.0026 |
| VCAM-1 (ng/mL) | 819 [623–959] | 2875 [1753–5108] | <0.0001 |
| E-Selectin (ng/mL) | 26.4 [22.5–33.7] | 56.7 [41.5–74.1] | 0.0001 |
| P-selectin (ng/mL) | 17.0 [15.4–20.6] | 22.2 [17.6–25.7] | **0.0621** |
| Angiopoietin-1 (ng/mL) | 0.4 [0.3–0.6] | 0.5 [0.2–0.9] | **0.8874** |
| Angiopoietin-2 (ng/mL) | 1.8 [1.5–2.1] | 4.3 [2.7–5.3] | 0.0003 |
| **Ang-2:Ang-1 ratio** | 4.2 [2.7–5.6] | 12.14 [2.7–40.2] | 0.03 |
| VWF-A2 (pg/mL) | 126 [120–150] | 218 [199–277] | <0.0001 |
| ADAMTS13 (ng/mL) | 1110 [483–1740] | 776 [572–1328] | 0.5485 |
| PAI-1 (pg/mL) | 78.9 [62.4–96.4] | 112 [69.3–242] | 0.1541 |
| CD40L (ng/mL) | 0.5 [0.4–0.7] | 1.0 [0.7–1.3] | 0.0001 |
| Syndecan-1 (ng/mL) | 1.8 [1.6–2.4] | 3.7 [2.9–6.0] | 0.0003 |
| IL-11 (ng/mL) | 3.5 [2.9–4.3] | 5.7 [4.7–6.4] | <0.0001 |
| TPO (ng/mL) | 2.0 [1.7–2.2] | 3.0 [2.6–3.4] | <0.0001 |
| CXCL4 (ng/mL) | 0.8 [0.6–1.2] | 1.4 [0.7–2.8] | 0.1236 |
| CXCL7 (ng/mL) | 0.4 [0.4–0.5] | 0.73 [0.4–1.7] | 0.1958 |
| SCF (pg/mL) | 47.61 [37.34–89.34] | 45.68 [36.22–61.39] | 0.1594 |

PvLDH: *P. vivax* lactate dehydrogenase.

* = under detection limit.

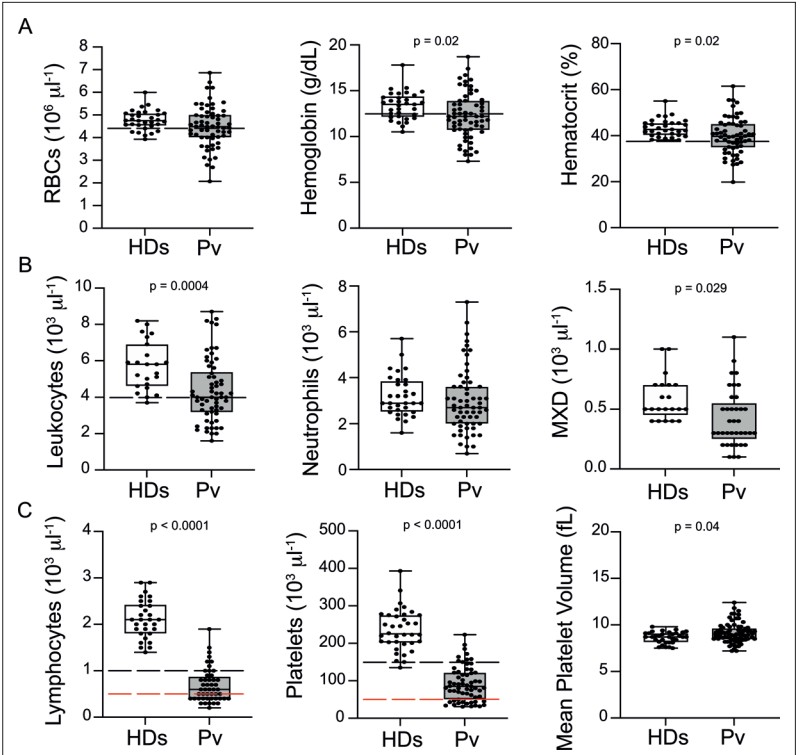

**Figure 1.** Clinical data of *P. vivax* patients (Pv) and healthy donors (HDs). (**A**) Red blood cell parameters. Shown are red blood cell counts, haemoglobin levels, and haematocrit. (**B**) Other blood cell parameters. Shown are numbers of leukocytes, neutrophils, and monocytes, basophils, and eosinophils (MXD). (**C**) Number of lymphocytes, platelets, and mean platelet volume (MPV). Parameters are depicted as box plots showing each individual value and the median with maximum and minimum values. Dashed lines in black mark the minimum threshold for normal reference values, while lines in red mark threshold for severe lymphopenia and thrombocytopenia, respectively. Two-tailed Student's t-test was used to compare variables with normally distributed data, and Mann–Whitney test was used to compare variables with non-normal distributions; p-value is indicated above the graph when p<0.05. HDs: healthy donors (controls, n = 34); Pv: *P. vivax*-infected patients (n = 79).

The online version of this article includes the following figure supplement(s) for figure 1:

**Figure supplement 1.** Demographic and clinical features of all *P. vivax*-infected patients compared with selected 31 patients for multiplex bead-based assay and downstream analysis.

(controls; *Table 1*). All individuals within the study including controls were from the state of Amazonas, in the Amazon region of Brazil. Blood was collected at enrolment for determination of haematological parameters, peripheral parasitaemia by Giemsa staining of blood smears, and PCR to determine genome copy numbers. Preparation of poor platelet plasma (PPP) was done within 15 min of sampling. The median peripheral parasitaemia was 4290 infected red blood cells (iRBCs)/μL of blood (25–75 interquartile range 1860–6620 parasites/μL) and parasite load of 26,642 copies of 18S RNA/ μL (25–75 interquartile range 9253–522,297). We also measured total parasite biomass independently of peripheral parasitaemia by quantifying levels of *P. vivax* lactate dehydrogenase (PvLDH) in plasma (*Table 1*).

Analysis of haematological parameters revealed significantly reduced haemoglobin levels and haematocrit across *P. vivax* patients compared to controls, with anaemia in 38% of the patients (*Figure 1A*). Similarly, leukocyte numbers were significantly decreased (mean ± SD: 4.36 ± 1.74 × 10³/ μL vs. 5.72 ± 1.34 × 10³/μL, p=0.0004), with 54.5% of the patients presenting with leukopenia (defined as a leukocyte count <4000 cells/μL). In contrast, neutrophil counts were not significantly different, and only 8.3% of *P. vivax* patients were presenting with neutropenia (neutrophil counts < 1500 cells/ μL) (*Figure 1B*). Other myeloid cell populations, however, such as monocytes, basophils, and eosinophils (MXD), were significantly reduced. We also observed a significant reduction in lymphocyte and platelet counts in this cohort (*Figure 1C*), with 80% presenting with lymphopenia (lymphocyte

counts < 1000 cells/μL) and 87% with thrombocytopenia (platelet counts < 150,000 cells/μL), many of them with severely reduced levels (*Figure 1C*). Alterations in platelet counts were accompanied by the release of mega platelets in the peripheral circulation as a significant increase on mean platelet volume was observed (*Figure 1C*).

In summary, patients in our cohort presented with a wide range of parasitaemia and uncomplicated clinical signs of *P. vivax* infection at medical consultation. However, significant haematological abnormalities were present in the majority of patients during early onset of disease, in line with previous findings (*Barber et al., 2015*; *Lacerda et al., 2011*; *de Mast et al., 2009*; *de Mast et al., 2007*; *Gomes et al., 2014*; *Park et al., 2003*; *Punnath et al., 2019*).

## Unsupervised clustering reveals two *P. vivax* patient subgroups that differ in parasite biomass: Vivax^high vs. Vivax^low

To determine whether the observed changes were associated with specific host signatures, particularly circulating biomarkers of haematological and endothelial changes, we applied a multiplexed microbead-based immunoassay (Luminex) in a representative subset of 31 *P. vivax* patients and 9 controls, as explained in the Materials and methods section (*Figure 1—figure supplement 1*). We selected a series of circulating biomarkers associated with haematological changes, including cytokines altering thrombopoiesis (TPO and IL-11), myelopoiesis, and lymphopoiesis (TNF-α, IL-1α, IL-1β, IL-6, IL-8, G-CSF) (*Boiko and Borghesi, 2012*; *Chiba et al., 2018*; *Kovtonyuk et al., 2016*). In addition, we selected markers of EC and platelet activation, coagulation (ICAM-1, VCAM-1, E-selectin, P-selectin, Angiopoietin-1 and -2, CD40L, VWF-A2, ADAMTS13, PAI-1, CXCL4, CXCL7), and EC glycocalyx breakdown (Syndecan-1).

We observed significant upregulation of multiple cytokines associated with haematological changes in the *P. vivax* patients compared to control (*Table 1*). In addition, patient samples exhibited a strong phenotype of increased EC activation, glycocalyx breakdown and coagulation. The high interquartile range in parasitaemia and host signatures (*Table 1*) suggested a heterogenous phenotype across the patient population. In order to identify possible stratification of patients into distinct subgroups, we further analysed the clinical data (*Figure 1*), parasite parameters, and Luminex data (*Table 1*) from the same 31 *P. vivax* patients and 9 controls as above. After z-score normalization, principal component analysis (PCA) was performed for data dimensionality reduction, considering the large number of variables in our dataset. Next, we ran K-means clustering (k) followed by bootstrapping (*Figure 2A and B*, *Figure 2—figure supplement 1*, *Figure 2—figure supplement 2*, *Figure 2—figure supplement 2—source data 1*) to identify possible subclusters of individuals. This analysis revealed consistent separation of samples into two clusters, one of them including all controls (cluster 1a ) and a subset of 14 patient samples (cluster 1b) and a second one representing the remaining 17 patient samples (cluster 2) (*Figure 2A and B*). In order to visualize covariables of the observed patient distribution (PCA) and clustering (K-means), we plotted the correlation (loading score) of each input variable with a principal component (PC; *Figure 2C*, *Figure 2—source data 1*). This analysis demonstrated covariation of lymphopenia and thrombocytopenia, on the one hand, and markers of EC changes, platelet production, activation, and parasite parameters (PvLDH and peripheral parasitaemia), on the other hand, as major contributors to the PCs (*Figure 2C*). Direct comparison of the two patient subgroups revealed significant higher total parasite biomass but not peripheral parasitaemia or parasite load (*Figure 3A*). In agreement with previous findings (*Barber et al., 2015*; *Fonseca et al., 2017*; *Silva-Filho et al., 2021*), z-score comparison further demonstrated that total parasite biomass was higher than and not correlated with peripheral parasitaemia levels or parasite load, particularly in patients of cluster 2 (*Figure 3B and C*). In addition, PvLDH was the input parasite variable with the highest loading score (correlation = 0.59) and lowest p-value (0.0000917) in the first PC dimension when compared with peripheral parasitaemia and parasite load (*Figure 2C*, *Figure 2—source data 1*). Indeed, using a best-fit classification tree model and a random forest machine learning model defining K-means clusters as categorical outcome, PvLDH is the best parasite predictor attribute segregating patients into clusters 1b and 2 (*Figure 3D and E*). After both models were trained in a set of 30 individuals, randomly selected by the training algorithm set, they were tested in the 10 remaining individuals, where all cluster 1a (control) individuals and 80% of *P. vivax* patients were correctly classified into either cluster 1b or cluster 2. Based on these observations, we designated cluster 1a as Control cluster (representing the healthy donors), cluster 1b as

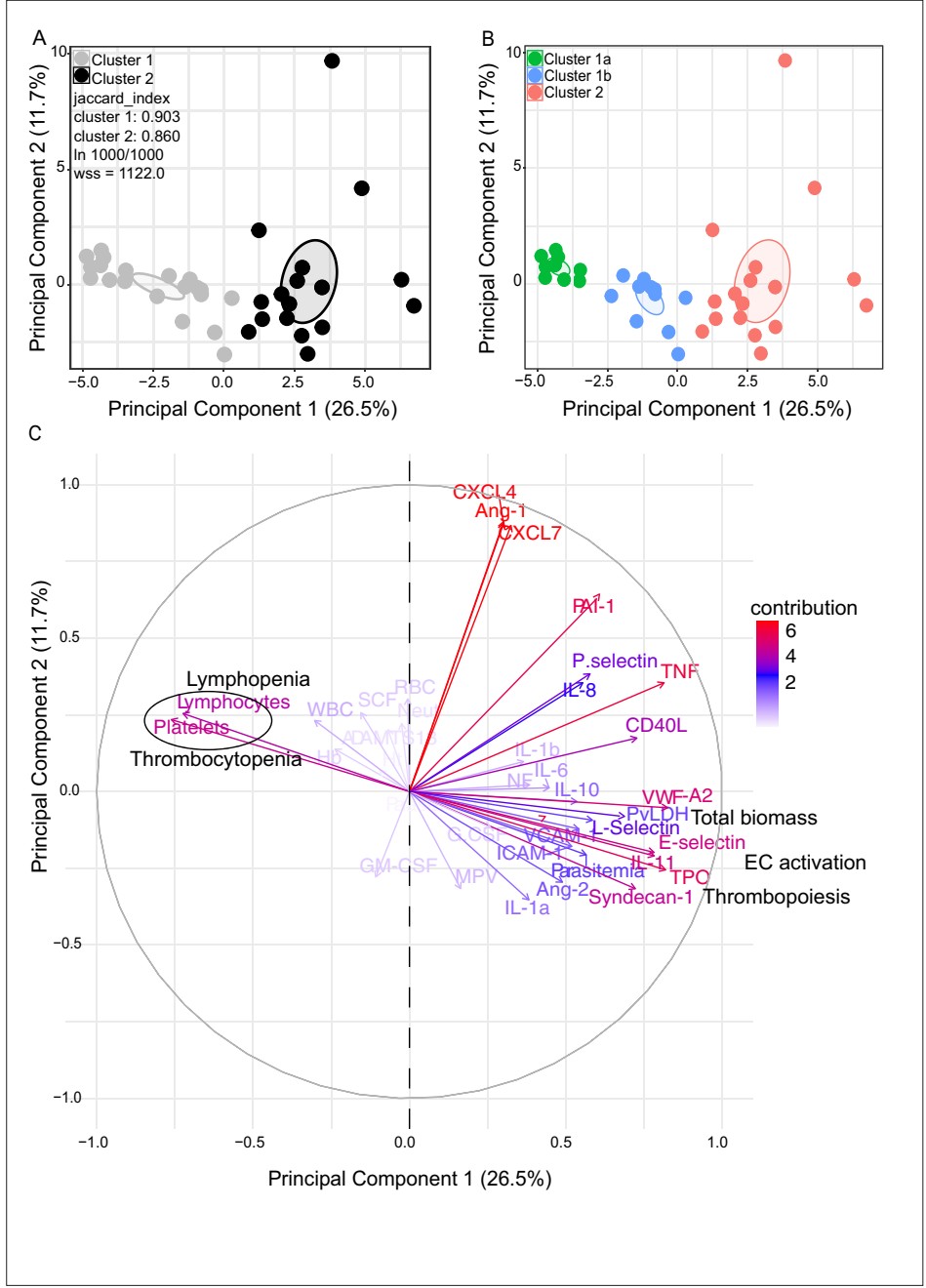

**Figure 2.** Characterization of heterogeneity in symptomatic *P. vivax* patients defines clusters of individuals. (**A, B**) Clustering of patients and healthy controls. After z-score normalization, principal component analysis (PCA) was performed for data dimensionality reduction. K-means clustering using k = 2 followed by bootstrapping (1000 times) in a PCA plot was performed and produced the most stable clusters regardless of the starting point (ln 1000/1000): cluster 1 = 23 individuals comprising 9 healthy donors and 14 *P. vivax* patients and cluster 2 comprising 17 *P. vivax* patients. The jaccard_index measures cluster similarity across bootstrap samples (jaccard_index ranges from 0 to 1, an index <0.6 hints at a weak, unreliable cluster while >0.85 means generally reliable). As indicated in the PCA plot, k = 2 gives stable clusters for all configurations (jaccard_index 0.9 and 0.86) and withinss (wss) = 1122. Open ovals represent 95% confidence interval ellipses around group mean points. PCA was performed for data dimensionality reduction, in parallel with K-means clustering (k) followed by bootstrapping (1000 times). Open ovals represent 95% confidence interval ellipses around group mean points. (**B**) The resulting clusters represent healthy controls (1a) and patients (1b, 2). (**C**) Contribution of variables to clustering. In the circular plot, the correlation between each input variable and principal components is used as coordinates (loading score). Plots show how covariables determine patient distribution in the PCA plot.

*Figure 2 continued on next page*

*Figure 2 continued*

The online version of this article includes the following figure supplement(s) for figure 2:

**Source data 1.** Correlation (loading score) of variables to principal components.

**Figure supplement 1.** Principal component analysis metrics.

**Figure supplement 2.** Methods determining the number of clusters best representing the data.

**Figure supplement 2—source data 1.** Measurements of K-means cluster stability, using k = 2, k = 3, and k = 4 clusters, via bootstrapping.

Vivax$^{low}$ (representing patients with low *P. vivax* biomass), and cluster two as Vivax$^{high}$ (representing patients with high *P. vivax* biomass).

## Different levels of haematological alterations between Vivax$^{high}$ and Vivax$^{low}$ patients

The three clusters were not significantly different in patient age (median: 33; IQ 25–75: 22–57), gender (80% male; 20% female in each cluster), average days of symptoms when samples were

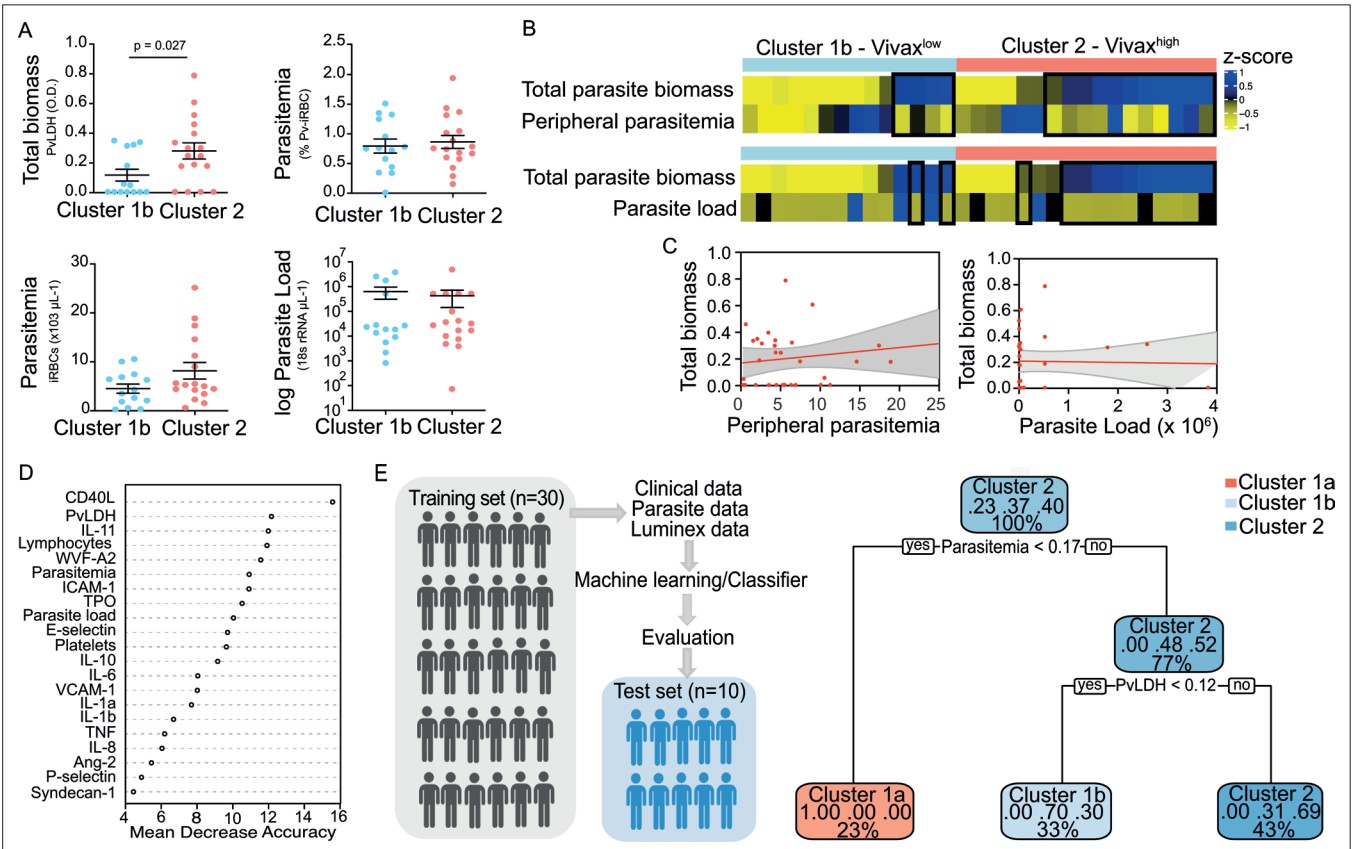

**Figure 3.** Unsupervised clustering analysis reveals two *P. vivax* patient subgroups that differ in parasite biomass. (**A**) Parasite parameters vs. patient clusters. Comparison of the two patient clusters (clusters 1b and 2) across parasite parameters reveals significant differences with total parasite biomass (*P. vivax* lactate dehydrogenase [PvLDH]) but not peripheral parasitaemia or parasite load (copies of 18S rRNA/µL of blood). (**B**) Parasite biomass vs. parasitaemia across clusters. Heatmap represents z-scores of PvLDH with peripheral parasitaemia or parasite load, respectively. Black boxes highlight patients with relatively lower peripheral parasitaemia compared to PvLDH levels, indicating the underestimation of total parasite biomass based on peripheral parasitaemia values. (**C**) Correlation between parasite biomass and parasitaemia. Scatter plot showing lack of correlation between PvLDH and peripheral parasitaemia or parasite load, respectively. Regression line in red, with 95% confidence interval shown in shaded grey. (**D, E**) Predicting parasite clusters. (**D**) Top parameters prioritized by random forest analysis ranked by the mean decrease in accuracy. (**E**) Best-fit decision trees and random forest machine learning models corroborate PvLDH value as the most important parasite signature in segregating patients into clusters 1b and 2. Cut-off values of the attribute that best divided groups were placed in the root of the tree according to the parameter value. The total of classified registers for each class and the percentage of observations used at that node are given in each terminal node.

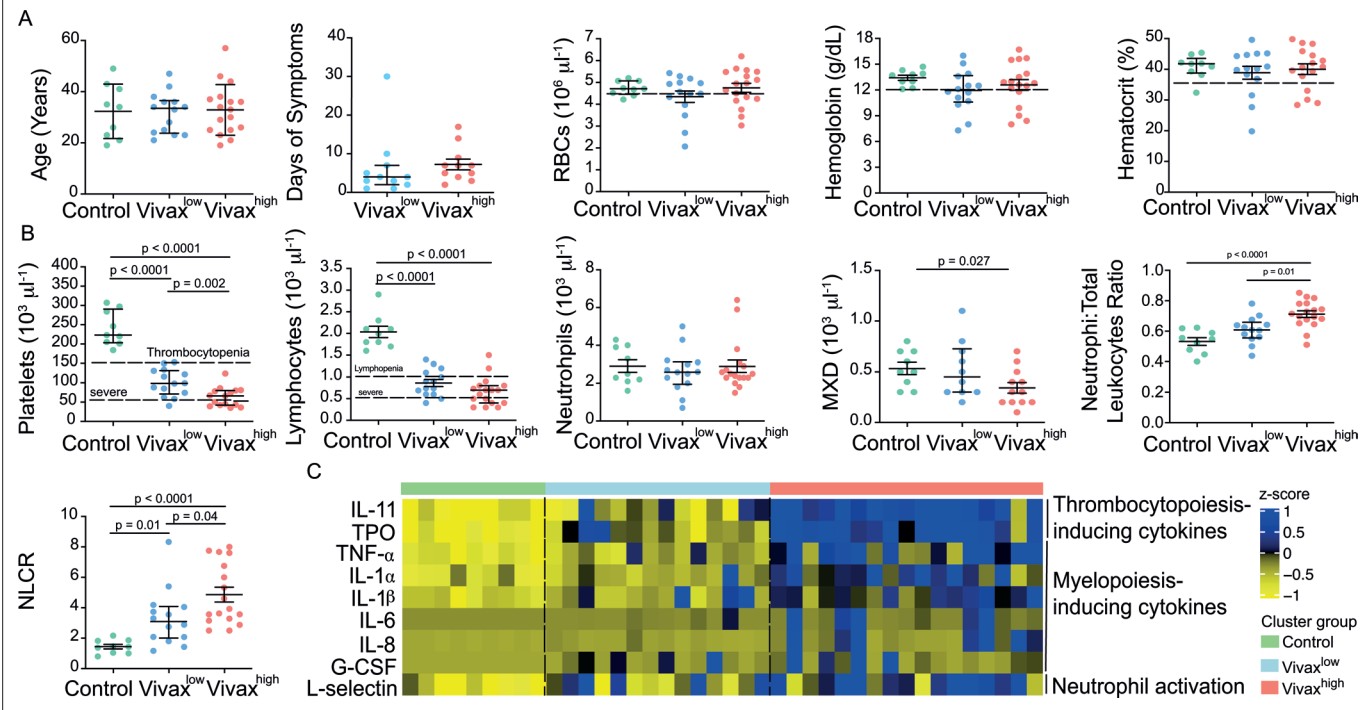

**Figure 4.** More severe haematological alterations in Vivax^high compared to Vivax^low patients. (**A**) Patient data and haematological parameters. Comparison of patient age, average days of symptoms when samples were collected, haemoglobin levels, haematocrit, or RBC counts across patient clusters (Control: n = 9; Vivax^low : n = 14; Vivax^high: n = 17). Data are depicted as plots showing individual values and the median (black lines) and the interquartile range. (**B**) Blood cell counts. Comparison of differential haematological counts across clusters. Shown are numbers of platelets, lymphocytes, neutrophils, and monocytes, basophils, and eosinophils (MXD), neutrophil to total leukocyte ratio, and neutrophil to lymphocyte ratio (NLCR). Top dashed lines mark the minimal threshold for normal reference values, while bottom dashed lines mark the threshold for severe lymphopenia and thrombocytopenia, respectively. Parameters are depicted as plots showing individual values and the median (black lines) and the interquartile range. One-way analysis of variance with Bonferroni-corrected multiple comparisons test was performed. p-Value is indicated above the graph when reached significance of p<0.05. (**C**) Cytokine response and neutrophil activation across clusters. Heatmap represents z-scores obtained by centering values of Luminex data. Shown are thrombopoiesis-inducing cytokines, myelopoiesis-inducing cytokines, and neutrophil activation markers. Biomarker concentrations were normalized (scale function in R), and the average scaled value is showed in blue and yellow scales. Blue shading represents the highest average scaled value, and yellow shading represents the lowest average scaled value. Each column (i.e. individual) in the heatmap is matched with colour-coded cluster assignment: Cluster Control – green bar; Cluster Vivax^low – blue bar; and Cluster Vivax^high – red bar.

The online version of this article includes the following figure supplement(s) for figure 4:

**Figure supplement 1.** Increase of thrombopoiesis- and myelopoiesis-inducing cytokines in the plasma of Vivax^high patients.

collected, haemoglobin levels, haematocrit, or RBC counts, indicating that these parameters are not confounders accounting for the differences observed between the clusters (***Figure 4A***). However, systematic analysis of haematological parameters between Vivax^high and Vivax^low patients revealed significant differences. Vivax^high patients showed a more intense reduction in platelet counts when compared to Vivax^low patients (Vivax^high 63,000 ± 6413 vs. Vivax^low: 100,700 ± 9381; p=0.002), with a higher frequency of patients with severe thrombocytopenia (Vivax^high 47% vs. Vivax^low 8%) (***Figure 4B***). Although not significant, there was a trend in the reduction of lymphocyte counts in Vivax^high patients when compared to Vivax^low, with 88% of Vivax^high patients presenting lymphopenia versus 64% in Vivax^low patients. In addition, we observed a fourfold increase in the frequency of patients with severe lymphopenia in the Vivax^high cluster compared to Vivax^low patients (***Figure 4B***). While there was no change in the number of circulating neutrophils in the different clusters of individuals, mixed cell counts (MXD), a parameter representing monocytes, basophils, and eosinophils numbers, were significantly reduced in Vivax^high patients. As a result, there was a significant enrichment of neutrophils in the leukocyte fraction in the blood of Vivax^high patients as well as a higher NLCR (***Figure 4B***).

In parallel to more severe thrombocytopenia in Vivax^high patients, plasma levels of cytokines inducing megakaryocytic differentiation in the BM, thrombopoietin (TPO), and IL-11 were significantly

increased in this cluster (*Figure 4C*, *Figure 4—figure supplement 1*). In accordance with the pattern of immune cell fractions in the peripheral blood of *P. vivax* patients, the Vivax[high] cluster showed a significant increase in the levels of proinflammatory cytokines associated with induction of myeloid-biased haematopoietic stem cell (HSC) differentiation and inhibition of lymphopoiesis in BM (e.g. TNF-α, IL-1α, IL-1β, IL-6, IL-8; *Figure 4C*, *Figure 4—figure supplement 1*; *Boiko and Borghesi, 2012*; *Chiba et al., 2018*; *Kovtonyuk et al., 2016*). In addition, Vivax[high] patients had increased circulating levels of G-CSF, a major mediator of HSC-biased myelopoiesis and the neutrophil activation marker, L-selectin (*Figure 4C*, *Figure 4—figure supplement 1*; *Soehnlein et al., 2017*; *Ivetic, 2018*; *Crockett-Torabi et al., 1995*). Together, these Luminex data support the haematological measurements, suggesting that a compensatory response is mounted in the BM to counterbalance the massive decrease of platelets in periphery. Upregulation of cytokines inducing myelopoiesis, while inhibiting lymphopoiesis (*Boiko and Borghesi, 2012*; *Chiba et al., 2018*; *Kovtonyuk et al., 2016*), might also explain the decrease of lymphocyte counts and enrichment of activated neutrophils in the peripheral circulation of *P. vivax* patients.

## Elevated circulating markers of EC activation and damage in Vivax[high] compared to Vivax[low] patients

Patient clustering indicated that Vivax[high] patients have increased levels of EC markers in the plasma compared to Vivax[low] patients (*Figure 2C*). Previous studies indicate that EC activation and damage might contribute to thrombocytopenia and inducing haematopoiesis, resulting in HSC differentiation directed towards myelopoiesis (*Lacerda et al., 2011*; *de Mast et al., 2009*; *de Mast et al., 2007*; *Boiko and Borghesi, 2012*; *Chiba et al., 2018*; *Kovtonyuk et al., 2016*; *Graham et al., 2016*; *Dos-Santos et al., 2020*; *Lazzari and Butler, 2018*). In our cohort, circulating levels of EC adhesion molecules (ICAM-1, VCAM-1, E-selectin, and P-selectin) and other EC activation markers and procoagulant molecules (Ang-2, VWF-A2, CD40L, and PAI-1) were significantly increased in the plasma of Vivax[high] patients compared to Vivax[low] patients and healthy controls (*Figure 5A*, *Figure 5—figure supplement 1A and B*). Likewise, Syndecan-1, a marker of EC glycocalyx breakdown (i.e. damage of EC plasma membrane; *Yeo et al., 2019*; *Pillinger and Kam, 2017*), was significantly increased in Vivax[high] but not in Vivax[low] patients (*Figure 5A*, *Figure 5—figure supplement 1C*).

To independently test whether host factors and/or parasite products present in the plasma of the different patient groups can directly induce changes in ECs, we stimulated primary human umbilical vein endothelial cells (HUVECs) with pools of plasma from either Vivax[high] patients, Vivax[low] patients, or healthy controls. These experiments demonstrated that only pooled plasma from Vivax[high] patients induces significant transcriptional upregulation of EC activation markers *ICAM-1*, *IL-1α*, and *IL-8* along with downregulation of *Ang-1*, *ADAMTS13*, and *NOS3* (eNOS) in HUVECs (*Figure 5B*, *Figure 5—figure supplement 1D*). In contrast, expression of *Syndecan-1* and *VEGF*, two indicators of vascular damage, was not affected by either treatment (*Figure 5B*, *Figure 5—figure supplement 1D*). Similarly, electric cell-substrate impedance sensing (ECIS) assays did not detect differences in functional perturbations in the endothelial cellular monolayer upon incubation with *P. vivax* pooled plasma when compared to control pooled plasma (*Figure 5C*). In contrast, flow cytometry and immunofluorescence assays performed with stimulated HUVECs revealed increased prevalence and protein expression levels of EC activation markers ICAM-1 and VCAM-1 upon exposure with Vivax[high] pooled plasma (*Figure 5D*, *Figure 5—figure supplement 1E*), in support of qRT-PCR data. These data indicate that local EC activation, mediated by direct or indirect interactions with parasitized RBCs, can be measured systemically.

## Indirect evidence for parasite-induced changes in deep tissues

To further investigate the interplay between host biomarkers and associated cellular responses as well as parasite parameters, we constructed a network of interactions based on Pearson's correlations with absolute correlation coefficient above 0.5 and p-value<0.05 (*Figure 6A*). In addition, we also performed hierarchical clustering on matrices of Pearson's correlations (p-value<0.01) with selected modules of parasite and host parameters (*Figure 6B*). Data from Vivax[low] and Vivax[high] patient subgroups were combined for this analysis as they similarly contribute to the associations we found so far (*Figure 6—figure supplement 1*).

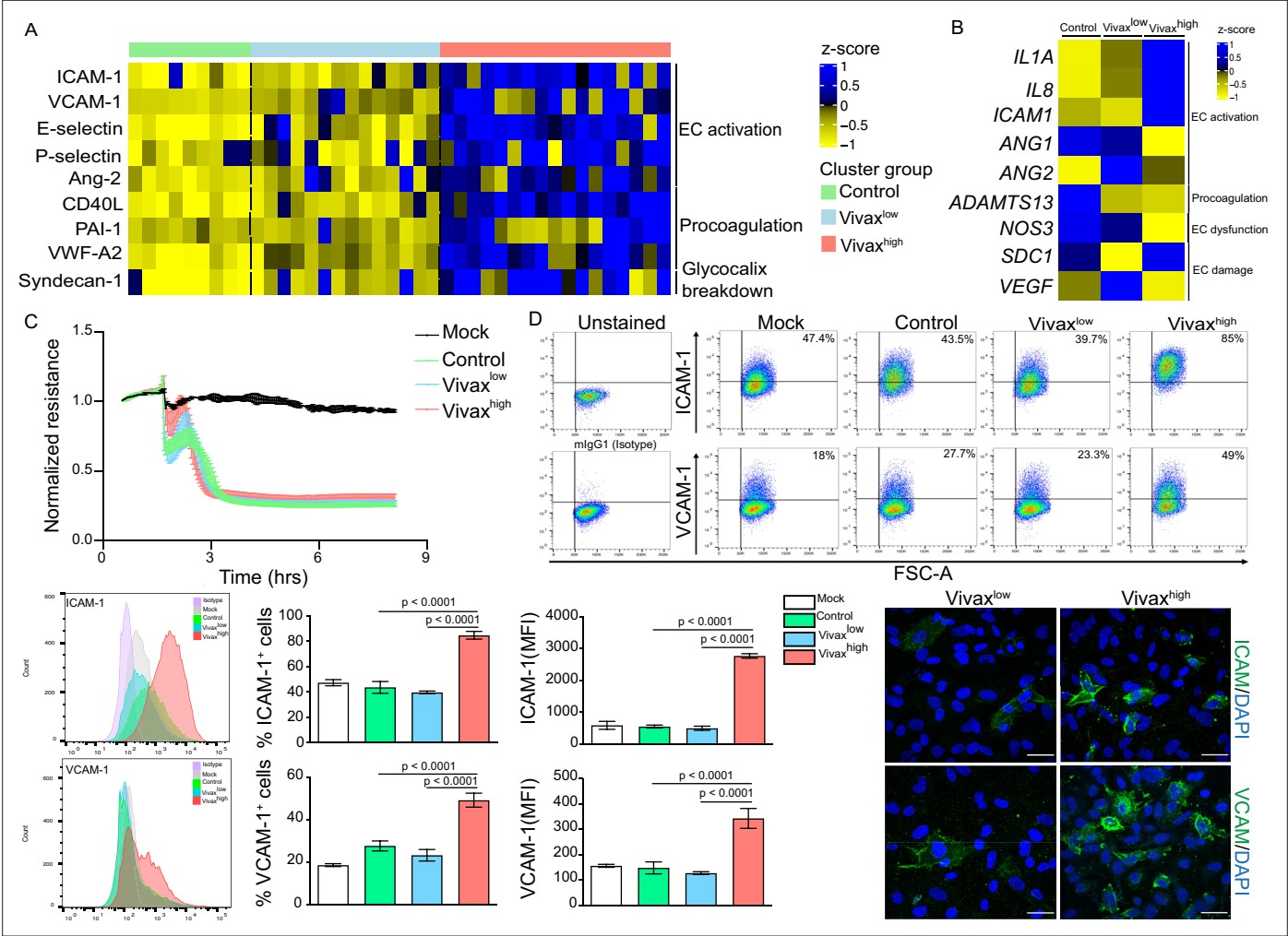

**Figure 5.** Elevated circulating markers of endothelial cell (EC) activation and damage in Vivax[high] compared to Vivax[low] patients. (**A**) Endothelial changes across clusters: Luminex. Heatmap represents z-scores obtained by centering values of Luminex data. Shown are markers of EC activation, procoagulation, and glycocalyx damage. Each column (each individual) in the heatmap is matched with colour-coded cluster assignment: Cluster Control – green bar; Cluster Vivax[low] – blue bar; and Cluster Vivax[high] – red bar. (**B**) Endothelial changes across clusters: qRT-PCR. Transcriptional response of human umbilical vein endothelial cells (HUVECs) incubated for 6 hr with 30% v/v pooled plasma from different clusters. Heatmap reflects relative mRNA expression intensity (average scaled value) after results were normalized to GAPDH housekeeping gene expression and untreated condition (mock). Data shown represent the mean of three independent experiments. For each experiment, two technical replicates were performed for each condition. (**C**) Endothelial changes across clusters: impedance changes. Endothelial monolayer integrity was measured during 20% v/v of pooled plasma incubation. Each line represents the mean ± SD of normalized resistance of HUVECs measured by electric cell-substrate impedance sensing (ECIS) at 4000 Hz. Data shown are representative of three independent experiments. For each experiment, two technical replicates were performed for each condition. (**D**) Endothelial changes across clusters: imaging and flow cytometry. HUVECs were incubated for 18 hr with 30% v/v of pooled plasma of individuals in the different clusters or left untreated (mock). Percentage of cells expressing EC activation markers (adhesion molecules) ICAM and VCAM as well as quantification of protein expression was determined by flow cytometry and immunofluorescence analysis (scale bar = 33 μM). Isotype antibodies were used as control to define positive populations. Significance was calculated for comparisons between conditions at the corresponding time point . One-way analysis of variance statistical test with Tukey's corrected multiple comparisons test was performed. p-Value is indicated above the graph when p<0.05. Data shown are representative mean ± SEM of three independent experiments.

The online version of this article includes the following figure supplement(s) for figure 5:

**Figure supplement 1.** Increase of markers of endothelial cell (EC) activation, damage (glycocalyx breakdown), and procoagulation in the plasma of Vivax[high] patients.

**Figure supplement 2.** Haemolysis potentiates Vivax[high]-induced endothelial cell (EC) activation.

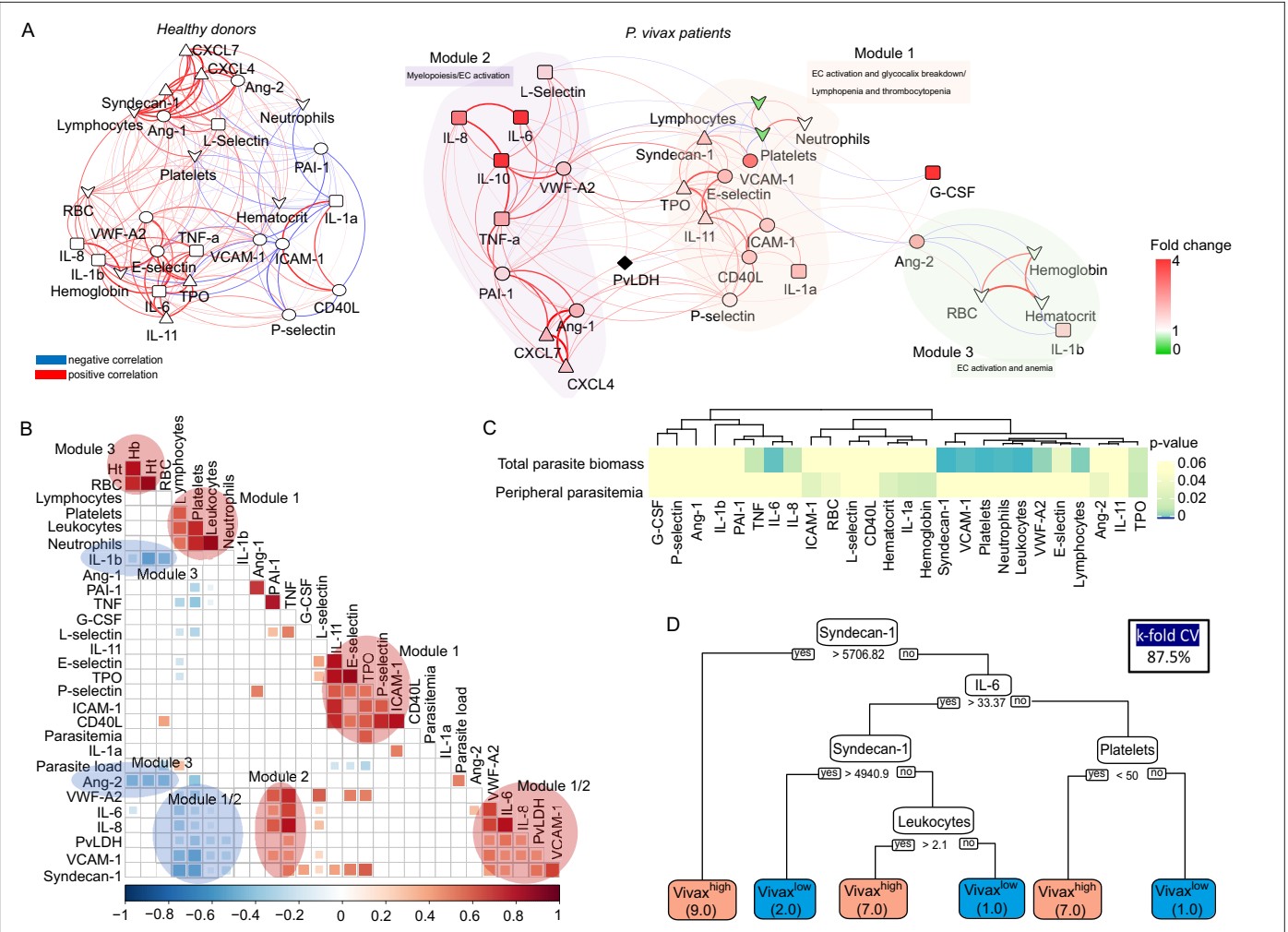

**Figure 6.** Network analysis and clustering of parasite and host signatures indicate parasite-induced changes in deep tissues. (**A**) Network analysis. Networks of the Pearson's correlations (absolute coefficient above 0.5 and p-value<0.05) between parasite biomass (*P. vivax* lactate dehydrogenase [PvLDH]) and host signatures in healthy donors (left graph) and in *P. vivax*-infected patients (right graph), using a force-directed layout. The symbols of the nodes represent biological functions: triangle represents markers of platelet activation and thrombopoiesis-inducing cytokines; V shape represents haematological parameters (neutrophil, lymphocyte, and platelet counts); circles represent endothelial cell activation markers; squares represent myelopoiesis-inducing cytokines and neutrophil activation markers. The colours in the nodes represent the fold change in relation to control levels. Because healthy donors do not have parasitaemia, PvLDH node is represented in black. Each connecting line (edge) represents a significant interaction detected by the network analysis using R. Correlation strength is represented by edge colour transparency and width. Positive correlations are represented by red edges, and negatives correlations are represented by blue edges. (**B, C**) Correlation matrix and heatmap. (**B**) Representative image of Pearson's correlation matrix calculated for all *P. vivax* patients. Only correlations with p-value<0.01 are shown, and hierarchical clustering was applied. Red circles highlight positive correlations in the functional modules depicted in (**A**), and blue circles highlight negative correlations in the functional modules also depicted in (**A**). (**C**) Heatmap showing p-values of the correlations between different parasite parameters, parasite biomass (PvLDH), and peripheral parasitaemia and host signatures (haematological and Luminex parameters). (**D**) Decision tree model. Best-fit classification tree model generated with the C4.5 algorithm showing Syndecan-1, IL-6, and platelet counts are the dominant variables capable of predicting total parasite biomass in *P. vivax* patients. Cut-off values of the attribute that best divided groups were placed in the root of the tree according to the parameter value (pg/mL for soluble markers or number of cells × 1000 /μL of blood for platelet counts). The total of classified registers for each class is given in parentheses for each terminal node with the k-fold cross-validation (k-fold CV) accuracy indicated.

The online version of this article includes the following figure supplement(s) for figure 6:

**Figure supplement 1.** Representative images of Pearson's correlation matrix calculated separately for each *P. vivax* patient cluster.

**Figure supplement 2.** Validation of patients' clusters and correlations when segregating patients based on thrombocytopenia severity.

**Figure supplement 3.** Validation of patients' clusters and correlations when segregating patients based on lymphopenia severity.

Similar to a previous study with *P. vivax* patients and healthy donors from an endemic area in Brazil (*Mendonça et al., 2013*), our analysis revealed a dense network of interactions with homogenous and centralized topology among the biomarkers in healthy donors (*Figure 6A*, *Supplementary file 1*). The network topology is drastically altered in symptomatic *P. vivax* patients, largely due to the introduction of parasite parameters in the patient graph (*Figure 6A*, *Supplementary file 1*). The network analysis revealed a decentralized topology, lower complexity and connectivity between the edges with data from *P. vivax* patients compared to the highly dense, homogenous and centralized network graph of healthy donors (91 edges vs. 166 edges, respectively). Of note, the network pattern described in our study is similar to protein-protein-associated networks described previously in *P. vivax* malaria and in other clinical contexts (*Mendonça et al., 2013*; *Frankenstein et al., 2006*). Interestingly, due to its decentralized and heterogenous patterns of interactions, the network graph of *P. vivax* patients is separated into three modules of strong interactions, with closely related biological functions. Module 1 is formed by markers of EC activation and damage, together with lymphocyte, platelet, and neutrophil counts in addition to the megakaryocyte differentiation-inducing cytokines (TPO and IL-11) (*Figure 6A*). In support of the role of EC activation and damage in the haematological changes observed in this cohort, hierarchical clustering revealed a positive correlation between adhesion molecules VCAM-1 and E-selectin and EC glycocalyx breakdown (Syndecan-1) (*Figure 6B*). In addition, VCAM-1, E-selectin, Ang-2 and VWF-A2, and Syndecan-1 are negatively correlated with platelet and lymphocyte counts, while ICAM-1 is positively correlated with neutrophil counts (*Figure 6A and B*). Module 2 is formed by proinflammatory cytokines with myelopoiesis-inducing effects and molecules associated with platelet activation and coagulation cascades (*Figure 6A and B*). Interestingly, EC activation markers and Syndecan-1 (EC damage) from module 1 also display positive correlations with myelopoiesis-inducing cytokines from module 2 (*Figure 6B*). Finally, module 3 is formed by Ang-2 and the proinflammatory cytokine IL-1β negatively associated with haemoglobin, haematocrit, and RBC numbers (anaemia markers) (*Figure 6A and B*). Most notably, PvLDH connects the two main functional modules 1 and 2 (*Figure 6A and B*). Accordingly with *Figures 2, 6A and B*, the biological significance of total parasite biomass, but not peripheral parasitaemia or parasite load, in affecting host response is also corroborated by the high significant and positive associations of PvLDH with multiple host parameters, including Syndecan-1 (EC damage), VCAM-1, VWF (EC activation and platelet pooling), and IL-6, IL-8, and TNF-α (inflammation and myelopoiesis-inducing cytokines) (*Figure 6C*). Meanwhile, platelet, lymphocyte, and neutrophil counts are negatively correlated with high significance (p-value<0.0001) with total parasite biomass, but not with peripheral parasitaemia or parasite load (*Figure 6C*). The association between endothelial activation, Syndecan-1, and parasite biomass (PvLDH) indicates a positive feedback loop between glycocalyx breakdown, activation of endothelial receptors such as ICAM-1 and VCAM-1, and parasite accumulation in deep tissues (*Carvalho et al., 2010*; *Lopes et al., 2014*). Similar to *Figure 2E*, application of a best-fit classification tree model demonstrated that Syndecan-1, IL-6, and platelet counts are the most dominant predictor attributes capable of classifying *P. vivax* patients based on total parasite biomass levels (*Figure 6D*). Using this model, all *P. vivax* patients were correctly classified into either low (Vivax[low]) or high (Vivax[high]) total parasite biomass (PvLDH). In turn, PvLDH is a relevant predictor attribute (high information gain) in predicting thrombocytopenia severity, and it is associated with increased severity of thrombocytopenia and lymphopenia in our cohort (*Figure 6—figure supplement 2*, *Figure 6—figure supplement 3*). Together, these data further support the hypothesis that a parasite population outside of circulation, as represented by total parasite biomass, is driving the host response including EC activation and damage as well as haematological disturbances (i.e. lymphopenia, thrombocytopenia, and anaemia) in *P. vivax* patients (*Figure 6—figure supplement 2*, *Figure 6—figure supplement 3*).

## Discussion

In this study, we performed a comprehensive analysis of host and parasite signatures detected in the plasma of a cross-sectional cohort of uncomplicated *P. vivax* malaria. Initial analysis of a series of circulating host biomarkers revealed significant levels of thrombocytopenia, lymphopenia, and anaemia, as well as EC activation and damage across *P. vivax* patients compared to healthy controls. Deconvolution of heterogeneity across patients revealed two patient subgroups (Vivax[high] and Vivax[low]) characterized by differences in total parasite biomass (based on circulating PvLDH levels) but not peripheral parasitaemia (based on blood smears). We observed a significant correlation between total

parasite biomass (but not peripheral parasitaemia) and systemic levels of markers of EC activation and damage and haematopoietic perturbations. In addition, by applying a supervised machine learning tree-structured model, we were able to associate EC damage and thrombocytopenia with parasite biomass. In agreement with a previous study (*Barber et al., 2015*; *Silva-Filho et al., 2021*), our observations further suggest that total parasite biomass as measured by PvLDH is a better predictor of *P. vivax* host responses and pathogenesis than peripheral parasitaemia. Furthermore, these findings support the emerging paradigm of a major *P. vivax* parasite reservoir outside of circulation, particularly in the haematopoietic niche of BM and spleen (*Silva-Filho et al., 2021*).

The existence of a significant *P. vivax* reservoir outside of circulation was first predicted by disproportionately high PvLDH levels in peripheral circulation compared to parasitaemia by blood smear (particularly in patients with complicated outcomes) and by modelling using experimental *Plasmodium cynomolgy* infections in NHPs (*Barber et al., 2015*; *Fonseca et al., 2017*). Recent studies provide direct evidence that BM and spleen represent the major reservoir of parasite biomass in *P. vivax* infection (*Obaldia et al., 2018*; *Baro et al., 2017*; *Brito et al., 2020*; *Kho et al., 2021a*; *Kho et al., 2021b*). PvLDH is produced by viable or recently killed parasites and hence considered a proxy for ongoing rather than past infection (*Barber et al., 2015*; *Druilhe et al., 2007*). PvLDH antigen capture ELISA established a direct relationship between pLDH levels and *P. vivax* parasitaemia in ex vivo experiments, demonstrating that pLDH reflects total *P. vivax* parasite biomass (*Druilhe et al., 2007*). Our study further explores the relevance of PvLDH as a prognostic marker of host perturbations and disease severity, with a particular focus on markers of changes in the haematopoietic niches of BM and spleen. A major observation in the network graph of *P. vivax* patients is the central position of the total parasite biomass marker PvLDH due to its equally strong interactions with the two main functional modules 1 and 2. Given that the haematopoietic niches of the BM and the spleen are the major reservoir of parasite biomass, interactions of PvLDH with these two main modules indicate an interplay between parasite infection in these niches and endothelial activation/damage as well as the proinflammatory response that results in myeloid-biased differentiation, thrombocytopenia, and lymphopenia. Furthermore, the highly significant and positive associations between endothelial activation, Syndecan-1, and parasite biomass (PvLDH) indicate a positive feedback loop between glycocalyx breakdown, activation of endothelial receptors such as ICAM-1 and VCAM-1, and parasite accumulation in deep tissues. Vivax[High] patients show higher plasma levels of all these markers. Consistent with previous reports (*Yeo et al., 2019*; *Barber et al., 2021*), we propose that elevated EC activation and glycocalyx damage increases the exposure of adhesion molecules, which in turn favours endothelial cytoadherence of *P. vivax*-infected RBCs, particularly in the splenic red pulp cords and in the BM (*Kho et al., 2021a*; *Introini et al., 2018*; *Hempel et al., 2017*; *Toda et al., 2020*). Accordingly, application of a best-fit classification tree model identifies Syndecan-1 as a putative host biomarker (EC glycocalyx breakdown marker) predicting total parasite biomass in *P. vivax* patients. We hypothesize that elevated endothelial activation and damage in Vivax[High] patients results in increased cytoadherence of *P. vivax* iRBCs and hence accumulation and growth in deep tissues, thus reducing the fraction of the parasite biomass in circulation.

In contrast to *P. falciparum*-infected individuals, a wide range of complicated clinical syndromes occur in *P. vivax* patients even at low or subpatent parasitaemia (*Baird, 2013*), thus indicating that peripheral parasitaemia is a poor predictor of clinical outcomes. Two lines of evidence support our conclusion that severity of infection is dependent on parasite biomass instead. First, the discrepancy between PvLDH levels and peripheral parasitaemia determined by blood smears is more evident in *P. vivax*-infected patients with complicated outcomes: the ratio of plasma pLDH to peripheral parasitaemia is sixfold higher than in non-severe patients. The same comparison between severe and non-severe *P. falciparum* patients reveals only a 1.4-fold difference (*Barber et al., 2015*). Second, although thrombocytopenia and lymphopenia are not included in the World Health Organization (WHO) criteria for defining severe malaria, it has been associated with severe manifestations and the need for blood and platelet transfusions in severe vivax malaria. This points out their clinical relevance in malaria diagnosis and treatment (*Lacerda et al., 2011*; *Naing and Whittaker, 2018*; *Gerardin et al., 2002*; *Kochar et al., 2010*; *Kochar et al., 2005*), suggesting that these haematological complications could be explored as markers of severity for this species. Both severe thrombocytopenia and lymphopenia were more frequent in patients in cluster 2 (Vivax[high]) in our study. By integrating these clinical perturbations with host biomarker measurements and parasite parameters, we demonstrated

the high attribute value of total parasite biomass in predicting the severity of thrombocytopenia and lymphopenia and highly significant correlations with endothelial activation, glycocalyx breakdown, and other markers of inflammation.

Thrombocytopenia, lymphopenia, and anaemia are the most frequent *P. vivax*- and *P. falciparum*-associated haematological complications (*Lacerda et al., 2011*; *Naing and Whittaker, 2018*; *Rodriguez-Morales et al., 2005*; *Tangpukdee et al., 2008*). In our cohort, 34, 85, and 87% of patients exhibited anaemia, lymphopenia, and thrombocytopenia, respectively. Various mechanisms have been proposed to explain the damage or excessive removal of platelets in *P. vivax* infection, including oxidative stress, platelet phagocytosis, IgG binding to platelet-bound malaria antigens, spleen pooling, or increased circulating nucleic acids levels (*Lacerda et al., 2011*; *Naing and Whittaker, 2018*; *Kochar et al., 2010*; *Andrade et al., 2010*). EC activation and damage also plays a role in intravascular platelet agglutination and increased platelet clearance from the circulation (*Park et al., 2003*; *Punnath et al., 2019*). Our data also demonstrate that thrombocytopenia is associated with an increase in IL-1, IL-6, IL-8, IL-10, and TNF-α. We also observed elevated levels of cytokines inducing megakaryocyte differentiation, TPO, and IL-11, suggesting that a compensatory response is mounted in the BM to counterbalance the massive decrease of platelets in the periphery. In contrast, the relatively large drop in peripheral lymphocyte numbers we observed in the *P. vivax* patients is likely non-specific effect, for example, pooling in the enlarged spleen rather than a response by *Plasmodium*-specific lymphocytes (*Hviid and Kemp, 2000*). Corroborating the potential role of total parasite biomass, rather than peripheral parasitaemia, in haematological disturbances (i.e. lymphopenia, thrombocytopenia, and anaemia), Figures S7 and S8 show that total parasite biomass increases accordingly with thrombocytopenia and lymphopenia severity. Patients with severe thrombocytopenia also show more severe leukopenia, lymphopenia, and mega platelets (higher MPV). In addition, plasma levels of cytokines – such as TNF-α, IL1-β, IL-8, IL-10; EC activation/damage markers, VCAM-1, E-selectin, VWF-A2, Ang-2, Ang-2:Ang1 ratio; Syndecan-1; thrombopoiesis-inducing cytokines, TPO and IL-11; platelet activation marker, CD40L; and neutrophil activation marker, L-selectin – follow the increase in thrombocytopenia severity (*Figure 6—figure supplement 2*). A similar pattern is observed when stratifying patients based on lymphopenia severity (*Figure 6—figure supplement 3*). Interestingly, a tree-structured model demonstrated that PvLDH, along with VCAM-1 and Syndecan-1, is a relevant predictor attribute (high information gain) in predicting thrombocytopenia severity in our cohort (*Figure 6—figure supplement 2*).

Our data support previous studies suggesting a role for EC activation and damage in increased leukocyte adhesion, intravascular platelet agglutination with increased platelet clearance from the circulation and skewing of haematopoiesis towards the myeloid lineage (likely at the expense of lymphopoiesis) in the BM (*de Mast et al., 2009*; *de Mast et al., 2007*; *Gomes et al., 2014*; *Boiko and Borghesi, 2012*; *Chiba et al., 2018*; *Kovtonyuk et al., 2016*; *Graham et al., 2016*; *Dos-Santos et al., 2020*; *Pillinger and Kam, 2017*). *P. vivax* elicits a stronger inflammatory response and more pronounced endothelial activation when compared with other *Plasmodium* infections with similar or higher peripheral parasitaemia (*Yeo et al., 2010*); however, the role of EC activation in *P. vivax* pathogenesis is not yet understood. Damage of the EC plasma membrane, as represented by glycocalyx breakdown, has been associated with poor prognostic outcome in *P. falciparum* (*Yeo et al., 2019*), but there is no data available for *P. vivax*. In our cohort, soluble EC activation biomarkers (e.g. ICAM-1, VCAM-1, E-selectin, Ang-2, CD40L, vWF-A2) and the EC damage product, Syndecan-1, are positively correlated with thrombocytopenia, lymphopenia, anaemia, and neutrophil enrichment in the peripheral blood. In addition, these biomarkers are positively correlated with increased circulating levels of cytokines inducing megakaryocyte differentiation (e.g. IL-11 and TPO) and with cytokines inducing myeloid-biased HSC differentiation (e.g. TNF-α, IL1-α, IL6, IL-8, and G-CSF), suggesting both direct and indirect links between EC activation and damage and haematological perturbations. Total parasite biomass-inducing EC activation might act synergistically with inflammatory changes potentially leading to splenic platelet pooling and platelet clumping in the vasculature without DIC (*Lacerda et al., 2008*; *Pillinger and Kam, 2017*; *Becker et al., 2015*). Likewise, increased activation-induced cell death (AICD) in T cells, splenic T-cell accumulation (*Hviid and Kemp, 2000*), or decreased lymphopoiesis due to myeloid-biased HSC differentiation induced by inflammatory cytokines and EC activation in the BM (*Boiko and Borghesi, 2012*; *Chiba et al., 2018*; *Silva-Filho et al., 2021*) might explain the severe lymphopenia and neutrophilia in vivax patients. Together, such mechanisms could explain

the link between parasite biomass and EC activation/damage with haematological changes observed in vivax patients that might contribute to pathogenesis and disease severity.

In a second series of experiments, we performed ex vivo stimulation of HUVECs with the plasma of the *P. vivax* cohort demonstrating that the mixture of parasite and host factors can directly induce EC activation in the absence of parasitized RBCs. Of note, functional differences between HUVECs and adult vascular endothelium, including lack of ABO blood group antigen expression, have been reported (*O'Donnell et al., 2000*; *Tan et al., 2004*). Hence, EC stimulation with patient plasma may be further evaluated using primary vascular ECs.

ECs are capable of responding to pathogens by sensing pathogen-associated molecular patterns (PAMPs) through pattern-recognition receptors (PRRs), which might play a key role in inducing EC activation when detecting *P. vivax* molecules enriched in the tissues where the parasite accumulates. ECs also express specific cytokine/chemokine receptors to detect proinflammatory signals released systemically or locally by activated immune cells in response to infection (*Bernardo et al., 2004*; *Bevilacqua, 1993*). As a result, EC activation induces exocytosis of secretory granules known as Weibel–Palade bodies that leads to the release of Ang-2 and VWF, as well as transcriptional programmes that activate expression of adhesion molecules such as ICAM-1, VCAM-1, E-selectin, and secreted cytokines and chemokines (*de Mast et al., 2007*; *Bernardo et al., 2004*; *Bevilacqua, 1993*). However, EC pathophysiology is complex, and changes represent a heterogenous spectrum ranging from simple perturbation to activation and even EC damage (*de Mast et al., 2009*). Our Luminex data clearly confirm such heterogeneity in the spectrum of EC changes due to *P. vivax* infection, with systemic increase of markers of EC activation and damage only detected in Vivax$^{high}$ patients. The ex vivo data show that increased systemic host proinflammatory factors and/or parasite products can alter EC properties, including activation of adhesion molecules and proinflammatory cytokines and downregulation of ADAMTS13. In contrast, vascular integrity was not affected. These data indicate that systemic inflammatory responses in *P. vivax* patients can lead to local EC activation but not vascular damage, central events in malaria pathogenesis. It is likely that other circulating factors that we have not directly measured in our study are also contributing to EC activation and vascular permeability. In particular, extracellular vesicles (EV) originating from ECs, platelets, and RBCs are present during malaria infection and are known to modulate the host immune response to the parasite (*Toda et al., 2020*; *Mantel et al., 2016*; *Mantel et al., 2013*). In *P. falciparum*, infected RBCs release EVs containing immunogenic parasite antigens, which activate macrophages, induce neutrophil migration, and alter endothelial barrier function (*Mantel et al., 2016*; *Mantel et al., 2013*). In *P. vivax*, plasma-derived EVs from iRBCs are taken up by human spleen fibroblasts (hSFs). This event signals NF-kB translocation and upregulation of ICAM-1 expression, facilitating cytoadherence of *P. vivax*-infected reticulocytes (*Toda et al., 2020*).

Although our study lacks longitudinal information, the findings might have clinical implications during and after treatment of vivax malaria. Several case reports demonstrate progressive clinical deterioration after commencement of treatment in *P. vivax* patients, associated with parasite killing that result in haemolysis of iRBCs and intravascular inflammation and oedema in response to the products released from these cells (*Anstey et al., 2007*; *Anstey et al., 2002*; *Tan et al., 2008*; *Val et al., 2017*). Patients presenting with a strong host response during acute infection might therefore be at increased risk of deteriorating and developing severe symptoms after commencement of treatment (*Figure 5—figure supplement 2*). Thus, identification of unique biological signatures in *P. vivax* patients might help to build rational approaches to the diagnosis, prognosis, and individualized treatment to modulate the host response to vivax malaria.

Altogether, our data indicate that changes in clinical parameters and biomarkers detected in the plasma of *P. vivax* patients are the result of both systemic host responses and local infection in tissue reservoirs such as BM and spleen. Our analysis shows that measuring a combination of host parameters (e.g. Syndecan-1, IL-6, platelet levels) and total parasite biomass (PvLDH) could predict the extent of parasite infection outside of circulation. Our data also instigate future investigations of systemic signatures with parallel analysis focused on tissue responses, particularly in reservoirs such as the haematopoietic niche of BM and spleen, which has great potential to advance better diagnosis and treatment of *P. vivax*.

## Materials and methods

| Reagent type (species) or resource | Designation | Source or reference | Identifiers | Additional information |
|---|---|---|---|---|
| Sequence-based reagent | qRT-PCR Oligonucleotides | This study | See *Supplementary file 2* | |
| Commercial assay or kit | Customized multiplex suspension detection system | R&D Systems | | |
| Commercial assay or kit | Accutase Cell Detachment Solution | BioLegend | Cat. #423201 | |
| Chemical compound, drug | Fixable Viability Dye eFluor 506 | ThermoFisher | Cat. #65-0866-14 | |
| Software, algorithm | FlowJo software (v10) | Ashland, OR | | https://www.flowjo.com |
| Software, algorithm | RStudio software (v1.4.1106) | RStudio, Boston, MA | | https://www.rstudio.com |
| Software, algorithm | Cytoscape software (v3.8.1) | NIGMS, Bethesda, MD | | https://cytoscape.org |
| Software, algorithm | GraphPad Prism 9 (v9.1.1 (223)) | GraphPad Software, San Diego, CA | | graphpad.com |
| Software, algorithm | ImageJ software | NIH, Bethesda, MD | | imagej.nih.gov |

## Patients

Peripheral blood and plasma samples were collected from 79 patients infected with *P. vivax*, as diagnosed by light microscopy, seen at FMT-HVD and 34 healthy donors (controls). Patients and healthy donors were age and sex-matched, with a frequency of 30% female and 70% male individuals in both groups. All individuals within the study were from a local vivax malaria epidemic area in the Amazon region of Brazil. All patients included were outpatients that did not meet WHO criteria for severe malaria. Diagnosis was further confirmed by quantitative PCR (qPCR) for both *P. vivax* and *P. falciparum*, using previously published nucleotide sequences (*Rosanas-Urgell et al., 2010*). Excluding other coinfections could have been of interest. However, the differential diagnosis for an acute febrile illness is very broad and it would be impractical to track all other possible diseases. In addition, the patients included in the present work had mild disease, and therefore were discharged from hospital after a positive malaria diagnosis. No further investigation on other infections was done. The main coinfection to be considered for an acute febrile illness with no localizing signs in our context is dengue fever. Although dengue coinfection in our cohort is possible, the incidence at the hospital is only 2.8% (*P. vivax*/dengue coinfection; *Magalhães et al., 2014Magalhães et al., 2014*). Thus, it is unlikely that such a coinfection would have a major impact on our findings. Exclusion criteria were (1) under 18 years of age, (2) pregnancy, (3) use of antimalarials, (4) chronic disease, (5) medication known to interfere with platelet count/function, and (6) smoking.

Anaemia is defined as haemoglobin <12.5 g/dL; haematocrit <37% ; RBCs counts <4.45 × $10^6$/µL. Thrombocytopenia is defined as a decrease in platelet counts to <150,000/µL. Based on platelet levels, patients were grouped into (1) non-thrombocytopenia (NT: platelet counts >150,000/µL), (2) mild thrombocytopenia (MT: platelet counts 100,000–150,000/µL), (3) moderate thrombocytopenia (MDT: platelet counts 50,000–100,000/µL), and (4) severe thrombocytopenia (ST: platelet counts <50,000/µL). Lymphopenia was defined as a lymphocyte count of less than 1000 cells/µL. Neutropenia was defined as a neutrophil count of less than 1500 cells/µL and neutrophilia as a neutrophil count of more than 7000 cells/µL *Punnath et al., 2019*; *van Wolfswinkel et al., 2017*.

## Preparation of poor platelet plasma

After signing the informed consent, 20 mL of venous blood were drawn by venepuncture in a syringe with 15% acid citrate dextrose as anticoagulant to minimize in vitro platelet activation. Complete blood counts were done within 15 min of blood sampling with a Sysmex KX21N counter. Whole blood was centrifuged at 180 g for 18 min at room temperature, without brake for gradient formation, to obtain the platelet-rich plasma (PRP). PRP was centrifuged at 100 g for 10 min for removal of residual leukocytes, and subsequently centrifuged at 800 g for 20 min to obtain the platelet pellet. Prostaglandin E1 at 300 nM was used to minimize platelet aggregation. The supernatant was centrifuged at 1000 g for 10 min to obtain platelet-poor plasma (PPP).

## Multiplex bead array assay

The biomarkers were analysed in thawed plasma with a customized multiplex suspension detection system (R&D Systems) for quantification of the following biomarkers:

(1) Proinflammatory and myelopoiesis-inducing cytokines: TNF-α, IL-1α, IL-1β, IL-6, IL-8, and G-CSF.
(2) EC activation and coagulation markers: ICAM-1, VCAM-1, E-selectin, P-selectin, Ang-1, Ang-2, von Willebrand factor (vWF-A2), CD40L, PAI-1, and ADAMTS13.

(3) Glycocalyx breakdown and EC damage marker: Syndecan-1
(4) Platelet activation markers: CXCL4 and CXCL7
(5) Megakaryocyte differentiation-inducing cytokines: TPO and IL-11; and other proteins such as IL-10, L-selectin, and SCF.

A representative set of 31 *P. vivax* patients were selected for the multiplex assay (*Figure 1—figure supplement 1*). These patients were selected to encompass the wide range of peripheral parasitaemia present in the cohort (260–25,150 infected RBCs/µL) and to match age, gender, and other haematological parameters to those that were not selected. Nine healthy donors matched for age and sex were also selected.

## PvLDH ELISA

To measure PvLDH in patient plasma samples, ELISA was performed using a matching pair of capture and detection antibodies (Vista Diagnostics International LLC, Greenbank, WA). Briefly, 96-well microtiter plate was coated with monoclonal anti-pLDH Vivax-specific (clone 3H8, Vista Diagnostics International LLC; RRID:AB_2892826) at a concentration of 1 µg/mL in PBS (pH 7.4) and incubated overnight at 4 °C. The plate was washed and incubated with blocking buffer (reagent diluent) at room temperature for 1 hr. After washing, samples were added and incubated for 2 hr. Next, plates were washed and biotinylated anti-PvLDH detection antibody (clone 6c9, Vista Diagnostics International LLC; RRID:AB_2892827), diluted 1:4000 in blocking buffer, was incubated for 2 hr at room temperature, followed by streptavidin-HRP incubation for 20 min at room temperature. Plates were washed and incubated for 20 min with substrate solution. Optical density was determined at 450 nm. Cut-off of positivity was defined by correcting absorbance values generated in the plasma samples from healthy donors (controls) by blank values (plate controls), with both values being in the same range. Absorbance values higher than controls were considered positive. In parallel, we used schizont extracts to perform standard curves and lower absorbance values were in the range of O.D = 0.03–0.04. All positive patient samples gave O.D. values equal to or higher than 0.05.

## PCA and K-means hierarchical clustering

Haematological parameters (haemoglobin levels, haematocrit, differential blood cell counts), parasite parameters (peripheral parasitaemia by blood smear, parasite load by qPCR, and parasite biomass PvLDH ELISA), and Luminex data (24 biomarkers) from the selected 9 healthy donors and 31 *P. vivax* patients were normalized to avoid variable-specific bias and z-score values were determined. Since the host response is complex and multidimensional (one dimension per Luminex biomarker), we applied dimension reduction and clustering for ease of downstream analysis. For this, all variables were used as input for PCA to reduce the dimensionality of data using the *PCA* function in the *FactoMineR* package in R. For visualization of PCA results, *ggplot2*, *factoextra,* and *corrplot* packages were used. For each PC, we determined which variables are better represented and the contribution (correlation or loading score) of each variable for each (PC). Investigation of eigenvalues and the percentage of explained variances retained by the PCs demonstrated that the first 10 PCs accounted for the variance of the data (*Figure 2—figure supplement 1*). However, variables were well represented by the first two PCs (Dim 1 and Dim 2), which were therefore retained for further analysis. In parallel, we performed K-means clustering (k) followed by bootstrapping, which produced the most stable clusters with k = 2 (cluster 1 = 21 individuals; cluster 2 = 18 individuals), which seemed to be the most consistent with the data (*Figure 2A*, *Figure 2—figure supplement 1*, *Figure 2—figure supplement 2*, *Figure 2—figure supplement 2—source data 1*). *Figure 2—figure supplement 2—source data 1* contains the numerical data representing cluster stability via bootstrapping. The metrics of interest is jaccard_index, which measures the cluster similarity across bootstrap samples. Similar to the above, k = 2 gives stable clusters for all configurations (jaccard_index 0.9 and 0.86). Using Monte Carlo reference-based consensus clustering (M3C) analysis (*M3C* function in the *M3C* package in R) indicated that k = 2 is the optimal number of clusters when using K-means clustering (*Figure 2—figure supplement 2C and D*), but when determining spectral clusters, different from elliptical k-means clusters, k = 3 gives the best number of clusters (*Figure 2—figure supplement 2E–G*).

dditional information

imary cells isolated from
e umbilical vein

ntibody clone MEM-111
(1:100)

ntibody clone 1.4C3
(1:500)
C (1:100)

(1:10)

ntibody clone HCD54
C (1:100)

ntibody clone 3h8
ISA (1 µg/mL)

ntibody clone 6c9
ISA (1:4000)

## Correlation plots and heatmap visualization

Heatmaps were created to visualize variable values using R function *Complex Heatmap*. They represent z-scores using row scaling obtained by centring represented variables with the *scale* function, followed by column clustering using average cluster method and Euclidean distance metric in R. The same software was used to determine pairwise Pearson's correlation coefficients between variables by running the function *cor* in the *ggcorrplot* package and visualized as a correlogram using R function *corrplot* in the *Hmisc* package displaying positive correlations in red and negative correlations in blue using p≤0.01 as a cut-off.

## Recursive partitioning decision tree classification and machine learning models

We used recursive partitioning decision tree classification models to evaluate dominant signatures (attributes) predicting a specific outcome. For decision tree construction, we applied the C4.5 algorithm, using the *RWeka*, *caret* (Classification and Regression Training) and *e1071* packages or the *rpart* package in R. First, the library *caret* is used to create a 10-fold training set to train the model. Then, the algorithm implements decision trees (using the J48 method, which is an open-source Java implementation of the C4.5 algorithm) starting with all instances in the same group, then repeatedly divides the data based on attributes until each item is classified. The attribute on which to divide is selected by information gain, a statistical technique for determining which attribute split will most cleanly divide the data. To avoid overfitting, sometimes the tree is pruned back. In parallel, the algorithm performs k-fold cross-validation to measure the performance of a given predictive model and indicates which one has the higher accuracy. Here, we used k = 10 to yield test error rate estimates that suffer neither from excessively high bias nor from very high variance (*James et al., 2013*). In parallel, features with mean decrease accuracy larger than six were used for random forest. In the random forest analysis, a thousand trees were built using R package *randomForest* (version 4.6.14). The normalized additive predicting probability was computed as the final predicting probability. Those selected important features were used for the random forest analysis on the test cohort for model validation.

## Stimulation of HUVEC with patients' plasma pools

After standardization procedures, primary HUVEC were stimulated or not (mock control) in culture media for 6 hr – to evaluate mRNA expression – or for 18 hr – to evaluate protein expression – with complete EGM-2 medium (Lonza) containing 30% (v/v) plasma pools generated from the three subgroups – healthy control, Vivax^low and Vivax^high – and 3 U/L heparin.

## Real-time quantitative RT-PCR

After 6 hr stimulation, total RNA was isolated from the cell lysate using the miRVana miRNA Extraction kit (Ambion) according to the instructions of the manufacturer. cDNA was synthesized with TaqMan Reverse Transcriptase (Applied Biosystems, Foster City, CA) and mRNA expression of genes were determined by qRT-PCR. Real-time qRT-PCR was performed on an ABI-Prism 7000 PCR cycler (Applied Biosystems) or on the CFX96 Real-Time PCR Detection System (Bio-Rad). Cycling parameters were 95 °C for 1 min and then 35 cycles of 95 °C (15 s) and 60 °C (1 min), followed by a melting curve analysis. The median cycle threshold ($C_t$) value and $2^{-\Delta\Delta Ct}$ method were used for relative quantification analysis, and all $C_t$ values were normalized to the GAPDH mRNA expression level. Results expressed as means and SEM of biological replicates are shown. The mock sample (HUVECs incubated with culture media only) was used as reference. The oligonucleotides used are described in *Supplementary file 2*.

## Endothelial cell flow cytometry (FC) and immunofluorescence analysis (IFA)

For IFA, cells were grown in eight-well chambered coverslips (IBIDI) until confluence. After 18 hr stimulation with plasma pools, cells were washed with PBS and fixed/permeabilized with ice-cold 100% methanol for 5 min at –20 °C. In brief, cells were incubated with 10% goat serum (ThermoFisher) to avoid secondary antibody nonspecific binding for 1 hr at room temperature and then incubated with specific primary antibodies to human ICAM-1 (mouse monoclonal clone MEM-111; Abcam; Cat. #

ab2213; RRID: AB_302892; used at a dilution of 1:100 in 10% goat serum); VCAM-1 (mouse mono-clonal clone 1.4C3; Abcam; Cat. # ab212937; RRID: AB_2892824; used at a dilution of 1:500 in 10% goat serum); and mouse IgG1 isotype control (Dako; Cat. # X0931; RRID: AB_2892825; used at a dilution of 1:10 in 10% goat serum) overnight at 4 °C. After washing, wells were overlaid for 1 hr with AF488-conjugated secondary antibody (used at a dilution of 1:500 in 10% goat serum) and Hoechst (diluted at 1:2000) at room temperature. For controls, primary antibodies were omitted from the staining procedure and were negative for any reactivity. The chambers were placed at 4 °C until use for immunofluorescence assay (IFA). Percentage of positive cells and expression profiles for ICAM-1and VCAM-1 were then determined using ImageJ software (NIH, Bethesda, MD).

In flow cytometry, after 18 hr stimulation with 30% plasma pools, cells were washed 2× with DPBS and treated with Accutase Cell Detachment Solution (BioLegend) at room temperature for up to 3 min or until the cells are detached. Cell count and viability with trypan blue dye were determined and cells were resuspended in ice-cold DPBS without calcium/magnesium, 0.5 mM EDTA, and 10% foetal bovine serum (FBS; Gibco). Cells were incubated with FcBlock (BD Biosciences, San Jose, CA), followed by incubation with unconjugated anti-VCAM (mouse monoclonal clone 1.4C3; Abcam; Cat. # ab212937) or AF488-conjugated anti-ICAM-1 (mouse monoclonal clone HCD54; BioLegend; Cat. # 322714; RRID:AB_535986) or unconjugated mouse IgG1 isotype control (Dako; Cat. # X0931) for 1 hr at 4 °C. Cells were then washed and incubated for 1 hr at 4 °C with secondary antibody AF488-conjugated anti-mouse IgG (ThermoFisher). Finally, cells were incubated with Fixable Viability Dye eFluor 506 (ThermoFisher) in DPBS without calcium/magnesium, 0.5 mM EDTA for 30 min at 4 °C. Cells were washed and resuspended in buffer and acquired using a BD FACSCelesta cytometer (100,000 events/sample). Percentage of positive cells and expression profiles for ICAM-1and VCAM-1 were then determined by the mean fluorescence intensity using FlowJo software (v10; Ashland, OR).

## Ex vivo evaluation of endothelial cell monolayer function

EC monolayer function was measured using ECIS, an electric cell-substrate impedance sensing system (ECIS Zθ, Applied Biophysics, Troy, NY), as previously described (*Santaterra et al., 2020*). The system then applies weak alternating currents through the electrode array and continuously measures the ability of the cell monolayer to impede the movement of electrons between adjacent EC (resistance). Briefly, cells were seeded at $2.5 \times 10^5$ cells/well on fibronectin-coated (10 µg/mL) eight-well arrays (8WE10, Applied Biophysics) containing interdigitated gold electrodes. ECs were seeded 48 hr before experiments and the resistance started to be recorded after 48 hr. Only wells with resistance >1500 ohms and stable impedance/resistance readings were used. Before stimulation, resistance was continuously monitored for 2 hr to confirm monolayer stability represented by a plateau in the resistance curve. Stimuli (20% v/v pooled plasma in complete medium) was then added to wells under continuous impedance/resistance monitoring for 12 hr. A baseline resistance value was recorded immediately prior to the addition of each stimuli, and results are expressed as a ratio from baseline resistance (normalized resistance).

## Network analysis

The values of each circulating factor measured in the plasma samples, as well as haematological parameters and parasite biomass in healthy donors and *P. vivax* malaria patients, were input in the RStudio software (version 1.4.1106, 2021) to determine pair-wise Pearson's correlation coefficients to generate correlation networks and the p-value to test for non-correlation was evaluated using p≤0.05 as a cut-off. In order to analyse the structure of the networks, edges list was generated in R using the functions *melt* (*reshape2* package), *graph_from_edgelist* (*igraph* package). Graphs were customized in the Cytoscape software (version 3.8.1) using the force-directed layout, which simulates a system of forces, determined by the correlation strength. In the equilibrium state, edges tend to have uniform length, and nodes that are not connected by an edge tend to be drawn further apart. Network topology and module analysis were performed using the NetworkAnalyzer, jActiveModules, and MCODE plugins in Cytoscape (*Cline et al., 2007*; *Doncheva et al., 2012*). *Supplementary file 1* shows the results for all parameters quantified in the comparative network topology analysis between the graphs for healthy donors and *P. vivax* patients.

## Statistical analysis

Fisher's exact test was used for categorical data. Data normality was checked by Shapiro–Wilk test. Student's t-test was used to compare means between groups with normally distributed data, and data sets with non-normal distributions were compared using Mann–Whitney test. All tests were performed two-sided using a nominal significance threshold of $p < 0.05$ unless otherwise specified. When appropriate to adjust for multiple hypothesis testing, Tukey's or Bonferroni corrected multiple comparisons test significance at the p-value $< 0.05$ threshold was performed unless otherwise specified. Data are presented as scatter plots with median and 25–75% interquartile range, box plots showing minimum to maximum range or means and SEM, unless otherwise stated. Analyses were performed and the graphs generated in GraphPad Prism 9 (version 9.1.1 [223], 2021) and RStudio software (version 1.4.1106; 2021). To ensure that differences observed between *P. vivax*- infected patients and controls, as well as between the clusters, were due to disease status and not confounded by age or sex, the clinical parameters were fitted as response variables in a linear model with sex and/or age fitted as explanatory variables. Age and sex were included in the model if their coefficients were estimated as different from zero with p-value $< 0.05$. The residuals from the linear model were then used as age- and/or sex-corrected parameters in subsequent analyses.

## Study approval

All subjects enrolled in the study were adults. Written informed consent was obtained from all participants, and the study was conducted according to the Declaration of Helsinki principles. The study was approved by the local Research Ethics Committee at Fundação de Medicina Tropical Dr. Heitor Vieira Dourado (FMT-HVD, Manaus, Brazil), under #CAAE: 54234216.1.0000.0005.

## Acknowledgements

We would like to thank all patients enrolled in this research and the support of the malaria diagnosis and field team at field team in the Fundação de Medicina Tropical Dr. Heitor Vieira Dourado (FMT-HVD) in Manaus, Brazil. The authors also gratefully acknowledge the help and assistance provided by the Central Laboratory of High-Performance Technologies (LaCTAD, University of Campinas) and the Institute of Infection, Immunity and Inflammation Flow Core Facility in the generation of some of the data reported in this manuscript. MM was supported by a Wolfson Merit Award from the Royal Society and Wellcome Trust Center award (number 104111). JLSF was supported by the Sao Paulo Research Foundation (FAPESP grant 2019/01578-2 and 2016/12855-9), and FTMC was supported by the Sao Paulo Research Foundation (FAPESP grant 2017/18611-7). MVGL and FTMC are CNPq research fellows.

## Additional information

### Funding

| Funder | Grant reference number | Author |
| --- | --- | --- |
| Fundação de Amparo à Pesquisa do Estado de São Paulo | 2019/01578-2 | João L Silva-Filho |
| Fundação de Amparo à Pesquisa do Estado de São Paulo | 2017/18611-7 | Fabio TM Costa |
| Wellcome Trust | 104111 | Matthias Marti |
| Fundação de Amparo à Pesquisa do Estado de São Paulo | 2016/12855-9 | João L Silva-Filho |

The funders had no role in study design, data collection and interpretation, or the decision to submit the work for publication.

## Author contributions
João L Silva-Filho, João CK Dos-Santos, Conceptualization, Data curation, Formal analysis, Investigation, Methodology, Project administration, Supervision, Validation, Visualization, Writing – original draft, Writing – review and editing; Carla Judice, Conceptualization, Data curation, Formal analysis, Investigation, Methodology, Validation; Dario Beraldi, Data curation, Formal analysis, Methodology, Software, Validation, Writing – review and editing; Kannan Venugopal, Formal analysis, Investigation, Methodology; Diogenes Lima, Data curation, Formal analysis, Investigation, Supervision, Validation; Helder I Nakaya, Conceptualization, Formal analysis, Supervision, Validation; Erich V De Paula, Conceptualization, Data curation, Formal analysis, Investigation, Methodology, Supervision, Validation, Writing – review and editing; Stefanie CP Lopes, Conceptualization, Data curation, Formal analysis, Funding acquisition, Investigation, Methodology, Supervision, Validation, Visualization, Writing – review and editing; Marcus VG Lacerda, Conceptualization, Data curation, Funding acquisition, Project administration, Supervision, Validation, Writing – review and editing; Matthias Marti, Formal analysis, Funding acquisition, Investigation, Methodology, Project administration, Resources, Supervision, Validation, Visualization, Writing – original draft, Writing – review and editing; Fabio TM Costa, Conceptualization, Data curation, Funding acquisition, Investigation, Methodology, Project administration, Resources, Supervision, Validation, Visualization, Writing – original draft, Writing – review and editing

## Author ORCIDs
João L Silva-Filho http://orcid.org/0000-0003-4762-2205
João CK Dos-Santos http://orcid.org/0000-0001-5916-9845
Carla Judice http://orcid.org/0000-0003-1839-053X
Helder I Nakaya http://orcid.org/0000-0001-5297-9108
Matthias Marti http://orcid.org/0000-0003-1040-9566
Fabio TM Costa http://orcid.org/0000-0001-9969-7300

## Ethics
All subjects enrolled in the study were adults. Written informed consent was obtained from all participants and the study was conducted according to the Declaration of Helsinki principles. The study was approved by the local Research Ethics Committee at Fundação de Medicina Tropical Dr. Heitor Vieira Dourado (FMT-HVD, Manaus, Brazil), under #CAAE: 54234216.1.0000.0005 and by the Research Ethics Committee at University of Campinas (UNICAMP, Campinas, Brazil), under #CAAE: 54234216.1.3001.5404.

## Decision letter and Author response
Decision letter https://doi.org/10.7554/eLife.71351.sa1
Author response https://doi.org/10.7554/eLife.71351.sa2

# Additional files

## Supplementary files
• Supplementary file 1. Topological analysis of the network graphs of healthy donors and *P. vivax* patients.
• Supplementary file 2. Oligonucleotides sequences used in the qRT-PCRs.
• Transparent reporting form

## Data availability
All data generated or analysed during this study are included in the manuscript and supporting files. Numerical tables and source data files have been provided. Table 1, Figure 2—source data 1 and Figure 2—figure supplement 2—source data 1 contain the numerical data used to generate the figures.

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
