## [Decision Letter]

[Editors' note: this paper was reviewed by Review Commons.]

**Acceptance summary:**

In contrast to *Plasmodium falciparum*, which is the deadliest of all malaria parasites, *Plasmodium vivax* which causes nearly 40% of malaria cases outside of the sub Saharan Africa has been the less studied parasite. The unique feature of *P. vivax* is its dormant phase, not detectable in the periphery, and which allows the parasite to emerge, causing a recurrent episode of malaria infection. In this study, the authors observed profound differences in immune responses amongst subjects with low versus high total biomass of *P. vivax*, which points to a reservoir of Plasmodium parasites outside of the peripheral circulatory system as having a profound affect on disease severity.

**Decision letter after peer review:**

Thank you for resubmitting your work entitled "Total parasite biomass but not peripheral parasitaemia is associated with endothelial and haematological perturbations in *Plasmodium vivax* patients" for further consideration by *eLife*. Your revised article has been evaluated by Dominique Soldati-Favre (Senior Editor), a Reviewing Editor and 1 reviewer. The following individual involved in review of your submission have agreed to reveal their identity: Rays Jiang (Reviewer #4).

The manuscript has been improved but there are some remaining issues that need to be addressed, as outlined below:

Please respond to the reviewers queries numbered 1 – 4.

*Reviewer #4:*

The manuscript by by Silva- Filho et al., entitled: "Total parasite biomass but not peripheral parasitaemia is associated with endothelial and haematological perturbations in *Plasmodium vivax* patients" reports the importance of total parasite mass of *P. vivax* infection, and its relationship with a variety of host hematological and immunological markers. The study classifies infection based on clinical features, parasitemia, and host features. I have the following questions:

1. The use of PvLDH in patient plasma as a proxy for parasite biomass has been established before. But the use of the biomass marker in combination with other host markers are novel. Can the authors explains more in-depth why the suspected deep tissue mass contribute to the PvLDH levels from different host tissues? Because the conclusion of this study relies on this parameter, an in-depth understanding of this marker is necessary.

2. One of the important conclusion is the severity of the infection is parasite-biomass dependent. While the current study does not include severe vivax cases, are there studies support the author's conclusion?

3. Results Page17. The authors found that the total parasite biomass was higher and not correlated with parasitemia, particularly in patients of cluster 2. Why is cluster 2 strong in this aspect?

4. Network and module analysis. The results seem to show that the healthy donor and the patients have different network structure and module numbers? Can the authors explain more on it?

---

## [Author Response]

Reviewer #2 (Evidence, reproducibility and clarity (Required)):This study focused on *P. vivax*, which is an important neglected human malaria killer. The reported evidence will have a significant impact on diagnosing infectious diseases. The language in the manuscript is very good. However, some typos were reported. Some paragraphs might need particular attention to punctuation. Overall, the work is very good. The statistics are straight forward. However, there are a couple of major points that must be addressed before publication. Some of my comments are just recommendations to clarify some sections of the text.Major comments:The statistical methods can be improved by using generalised mixed models (GLMM).1. PCA graphs need to be organised in more descriptive ways. Dim1 and Dim2 in each axis need to be defined clearly in the figures. PCA in Figure 2 c is very difficult to follow, and it needs to be organised.

Figures have been amended to be more self-explanatory and clearer to the reader.

2. In this study, patients were male and female, and we know already male and female haematological parameters are hugely different, specially Hb level, and so on. My question is how the sex variable is treated in this study? Did your control group were from both sexes? Sex could be treated as a random variable in all studies if GLMM models were used.

Information in how the sex variable was treated in the study has been added to the methods section. In our cross-sectional study with uncomplicated *P. vivax* malaria patients seen at FMT-HVD in Manaus, Brazil, patients and healthy donors (controls) were matched by age and sex. In both groups, frequency of female individuals was 30% and male individuals 70%.

We think sex is better fitted as fixed effect since only two levels for this factor are possible. Thus, we used linear models with age and sex as fixed variables for statistical testing and to ensure that the differences observed between *P.vivax*- infected patients and controls, as well as between the clusters, were only due to disease status. This analysis showed that red blood cells count, hemoglobin, hematocrit, MXD and neutrophils counts (this parameter only when comparing the clusters) needed to be corrected only due to sex influence. For these parameters, estimates of predicted sex influence were subtracted from the raw parameter values and residuals were used for statistical testing. We have added this information in the Methods section as indicated below:

“Patients and healthy donors were age and sex-matched, with a frequency of 30% female and 70% male individuals in both groups.”

“To ensure that differences observed between *P. vivax* – infected patients and controls, as well as between the clusters, were due to disease status and not confounded by age or sex, the clinical parameters were fitted as response variables in a linear model with sex and/or age fitted as explanatory variables. Age and sex were included in the model if their coefficients were estimated as different from zero with p-value < 0.05.”

The residuals from the linear model were then used as age and/or sex corrected parameters in subsequent analyses.

3. Why 6h and 18h used for the HUVEC evaluation?

We ran several optimization experiments with individual plasma samples where we observed maximal mRNA expression changes after 6h of stimulation. For experiments detecting protein expression (IFA and flow cytometry), we increased the stimulation time to 18h. Preliminary experiments suggested this to be the optimal duration without compromising cellular viability.

4. It is mentioned only neutrophil enriched in this study, if myelopoiesis is affected, why the other granulocytes were not showed significant enhancement?

Our data reveal no change in the number of circulating neutrophils in the different clusters of individuals. However, mixed cell counts (MXD), a parameter representing monocytes, basophils and eosinophils numbers, was significantly reduced in Vivax^high^ patients. As a result, there was a significant enrichment of neutrophils in the leukocyte fraction in the blood of Vivax^high^ patients as well as a higher Neutrophil:Lymphocyte count ratio (NLCR) (Figure 4). In hematopoietic progenitors, stochastic changes in each factor’s concentration could result in one factor’s becoming more abundant and committing a hematopoietic progenitor to a particular lineage. To generate each mature granulocyte population (e.g. basophils, eosinophils and neutrophils), common myeloid precursor cells (CMPs) and later precursors for granulocytic and monocytic lineages (GMPs) follow in the BM different lineage commitment programs, tightly-regulated or instructed by a specific set of soluble factors, cell-cell interactions and transcription factors, that define cell fate decisions and lineage restrictions. For instance, differential PU.1 activity can specify different cell fates during haematopoiesis regulating monocyte and neutrophils differentiation. Genetic and biochemical analyses have shown that G-CSF can direct granulocyte differentiation by changing the ratio of C/EBPα to PU.1 (Zhu et al., Oncogene 2002; Friedman Oncogene 2002; Dahl et al., Nat Immunol 2003). High expression levels of PU.1 and C/EBPα, stimulated by G-CSF, promote GMP differentiation to neutrophils and inhibits monocyte differentiation, while only PU.1 expression, IRF-8 and lower expression/activity of C/EBPs induce GMP differentiation to monocytes (Zhu et al., Oncogene 2002; Friedman Oncogene 2002; Dahl et al., Nat Immunol 2003). Meanwhile, a combination of PU.1, C/EBPβ and low levels of GATA-1 differentiates GMPs to eosinophil lineage (Kulessa et al., 1995; McDevitt et al., 1997; Yamaguchi et al., 1999) and PU.1 must also cooperate with GATA2 to direct mast cell differentiation (Walsh et al., Immunity 2002). In addition, eosinophil and basophil differentiation are induced by a different set of cytokines, usually produced in prevalent T-helper 2 response, such as IL-5, which should be inhibited in the strong Th1 environment evidenced by our and previous Luminex data in Pv patients. The enrichment of activated neutrophils in the peripheral circulation of *P. vivax* patients could be due to a response that specifically enhances neutrophil production and release from the bone marrow (BM). This hypothesis is supported by emerging evidence for enrichment of *P. vivax* parasites in the hematopoietic niche of BM, our Luminex data showing significant increase in pro-inflammatory cytokines associated with emergency myelopoiesis (e.g., TNF-α, IL-1α, IL-1β, IL-6, IL-8), and increased circulating levels of G-CSF, the major inducer of neutrophils production in the BM. Likewise, increased activation-induced cell death (AICD) in T cells, splenic T-cell and platelet accumulation or decreased lymphopoiesis due to myeloid-biased HSC differentiation induced by inflammatory cytokines and EC activation in the BM (refs 36,37,39) might explain the neutrophil enrichment in vivax patients.

5. I would also ask the authors to speculate a bit on, What could be the mechanism behind the different function of *P. vivax* compared to *P. falciparum*? From an evolutionary perspective, the parasite should rather become softer and keep the host alive for its own benefit.

One of the characteristics of *P. vivax* that could play an important role in immunity is its restriction to invade immature reticulocytes. For example, the infected reticulocyte could play a role in the presentation of parasite antigens as reticulocytes (but not mature RBCs) express MHC-I and are capable to process and present antigens on their surface for recognition by T cells. Indeed, it has been shown that reticulocytes act directly as an antigen-presenting cell, emphasizing the importance of erythrocyte surface antigens both in the induction as well as the target of a protective immune response (Burel et al. 2016, Junqueira et al. 2018). Recent investigations comparing *P. vivax* and *P. falciparum* controlled human infection models (CHMIs) also revealed marked differences in the immune profiles generated following infection with the two species and postulated that protective immune responses to *Plasmodium* are species-specific. It has been hypothesized that this difference is due to strict *P. vivax* tropism for MHC-I-expressing reticulocytes that, unlike mature red blood cells, can present antigen directly to CD8^+^T cells. Specifically, *P. vivax* but not *P. falciparum* infection led to the expansion of a specific subset of CD38^+^CD8^+^ T cells which were associated with an activated phenotype and cytotoxic potential. Corroborating Burel et al. findings in the CHMI model, Junqueira et al. showed that *P. vivax*–infected reticulocytes express HLA-I. In *P. vivax*-infected patients, CD8^+^ T cells in the peripheral blood express high levels of cytotoxic proteins, recognize and form immunological synapses with *P. vivax*–infected reticulocytes in HLA–dependent manner. Next, it was showed that *P. vivax*-specific CD8^+^ T cells release their cytotoxic granules to kill both host cell and intracellular parasite, which prevented reinvasion (Junqueira et al. 2018). Although these data indicate a protective role of cytotoxic CD8^+^ T cells during *P. vivax* blood-stage malaria, it is not clear whether these lymphocytes would always be beneficial because they might contribute to anemia, inflammation or other pathological sequelae of infection, which needs to be further investigated.

Minor comments:

1. It is important to have a reference, version, and date for the R software, packages and GraphPad.

We have added version and date for the R and GraphPad software.

2. In Figure 5, E missed to report. This figure can be better organised. It is very hard to read and follow.

There is no E in Figure 5. We will organize the figure to make it easier to read and follow.

Reviewer #2 (Significance (Required)):*P. vivax* remains endemic in 51 countries across Central and South Americas, the Horn of Africa, Asia and the Pacific islands. In most areas it is co-endemic with *P. falciparum*, which has been the priority species to address for national malaria control programmes. Malaria related deaths are mostly attributable to the more pathogenic *P. falciparum*, but over the last decade these have declined, however there has been a consistent rise in the proportion of malaria cases due to *P. vivax*. However, because it is difficult to diagnose resistant strains, strategies to detect and track drug resistant *P. vivax* are limited. In this context it is vital to develop better tools to assess diagnostic, antimalarial efficacy and drug susceptibility so that emerging drug resistance can be tracked, and novel treatment strategies explored.From my viewpoint, despite some statistical problems to understand the complex nature of data (mixed interactions among multiple variables), these findings seem to be very interesting and (after a major revision) worth to be published. As said before, the story told by the authors could become interesting.Reviewer #3 (Evidence, reproducibility and clarity (Required)):The manuscript titled: "Total parasite biomass but not peripheral parasitaemia is associated with endothelial and haematological perturbations in Plasmodium vivax patients" by Silva-Filho et al., reinforce the original observation and data by the group of Nicholas Anstey and coworkers, who first proposed the use of plasma parasite lactate dehydrogenase and PvLDH as a marker of parasite biomass. In that work, it was already demonstrated that *P. vivax* biomass is related to plasma concentration of LDH levels. As such, the present work cannot be considered of high novelty. Yet, through a meticulous approach including clinical data, computational approaches, machine learning, LDH measurement, multiplex analysis and quantitave RT-PCR, the authors here have extended the original observations that a large biomass of *P. vivax* parasites is out of blood circulation. In contrast, unlike the original observations of Anstey´s group, a correlation between total parasite biomass and systemic levels of markers of endothelial cells activation, was observed. The manuscript is very well written and the discussion brings new knowledge in this key topic for elimination of malaria. This manuscript is therefore recommended for publication after the following comments are addressed.Major comments:1. The vascular endothelium plays a pivotal role in malaria. Therefore, to test whether cell and/or parasite factors affect the vascular endothelium, HUVEC cells were used in this study. This is of major concern as endothelial cells from the bone marrow, where most hematological disturbances, notoriously thrombocytopenia, occur, were not used instead. HUVEC cells seems the only endothelial cell that does not express ABO blood group antigens, thus suggesting that surface expression on these cells is highly altered (O´Donnell et al., 2000 J Vasc Res). Moreover, significant functional differences between HUVEC cells and adult vascular endothelium have been reported (Chan et al., 2004). Together, this indicates that results obtained with HUVEC cells might not reflect responses of the bone marrow vascular endothelium. As one of the corresponding authors have ample experience with working with human bone marrow endothelial cells (Mantel et al., 2016 Nat Comm), it is suggested to perform some experiments with these cells to assure extrapolation of the results obtained with HUVEC cells.

We agree with the reviewer that performing ex vivo assays with primary human bone marrow endothelial cells would be an excellent alternative. However, we would like to argue that HUVECs are also suitable for our purposes. HUVECs are widely used to study endothelial barrier function, for example in the context of angiogenesis and inflammatory responses/barrier disruption. To emphasise this point, we have now referenced examples where HUVECs were used in the context of endothelial barrier biology and in different inflammatory conditions (see also lists a, b, c below).

A) Papers showing the use of HUVECs in studies yielding important insights about endothelial barrier function

– Krispin S et al. Growth Differentiation Factor 6 Promotes Vascular Stability by Restraining Vascular Endothelial Growth Factor Signaling. Arterioscler Thromb Vasc Biol. 2018.

– Aranda JF et al. MYADM controls endothelial barrier function through ERM-dependent regulation of ICAM-1 expression. Mol Biol Cell. 2013.

– Orsenigo F et al. Phosphorylation of VE-cadherin is modulated by haemodynamic forces and contributes to the regulation of vascular permeability in vivo. Nat Commun. 2012.

B) Papers that used HUVECs in studies about endothelial barrier function in inflammatory conditions

– Dickinson CM et al. Leukadherin-1 ameliorates endothelial barrier damage mediated by neutrophils from critically ill patients. J Intensive Care. 2018.

– Kuck JL et al. Ascorbic acid attenuates endothelial permeability triggered by cell-free hemoglobin. Biochem Biophys Res Commun. 2018.

– Tramontini Gomes de Sousa Cardozo F et al. Serum from dengue virus-infected patients with and without plasma leakage differentially affects endothelial cells barrier function in vitro. PLoS One. 2017.

– Fox ED et al. Neutrophils from critically ill septic patients mediate profound loss of endothelial barrier integrity. Crit Care. 2013.

– Rahbar E et al. Endothelial glycocalyx shedding and vascular permeability in severely injured trauma patients. J Transl Med. 2015.

C) Papers showing that HUVECs behave similarly to other endothelial cell types in regard to barrier function, except when the comparison is with blood brain barrier models

– Totani L et al. Mechanisms of endothelial cell dysfunction in cystic fibrosis. Biochim Biophys Acta Mol Basis Dis. 2017, Dec;1863(12):3243-3253.

– Gündüz D et al. Effect of ticagrelor on endothelial calcium signalling and barrier function. Thromb Haemost. 2017 Jan 26;117(2):371-381.

– Deitch EA et al. Mesenteric lymph from rats subjected to trauma-hemorrhagic shock are injurious to rat pulmonary microvascular endothelial cells as well as human umbilical vein endothelial cells. Shock. 2001 Oct;16(4):290-3.

Importantly, we were able to reproduce in the HUVEC ex vivo assays a phenotype of endothelial perturbations that is inferred based on the in vivo Luminex data using the same plasma sample. These data also support our hypothesis that patients with higher parasite biomass present higher endothelial cell perturbations, corroborating the associations between parasite accumulation in deep tissues (total parasite biomass represented by PvLDH levels) and endothelial cell activation as demonstrated in the Figure 6.

2. Strikingly, the authors stated that "*P. vivax* infection results in different ranges of EC alterations without massive cytoadhesion". This statement has no data supporting it. In fact, their own flow cytometry data convincingly demonstrated that exposure of HUVEC cells to plasma of vivax-high patients significantly increased the surface expression of ICAM-1 and VCAM. ICAM-1 expression is a well know receptor for cytoadhesion in malaria and Dr. Costa first demonstrated the importance of this receptor in cytoadherence of *P. vivax* (Carvalho et al., 2010). Moreover, these data are in some contradiction with the original observations of Anstey and collaborators who demonstrated that parasite LDH concentration did not correlate with markers of endothelial activation (Barber et al., 2015 PLoS Path). Therefore, this sentence should be modified to accommodate the alternative possibility of cytoadherence, deleted from the manuscript or binding functional assays should be performed to sustain it.

We agree with the reviewer and have removed this statement.

“The association between endothelial activation, Syndecan-1 and parasite biomass (PvLDH) indicates a positive feedback loop between glycocalyx breakdown, activation of endothelial receptors such as ICAM-1and VCAM-1 and parasite accumulation in deep tissues^9,12^.”

3. Extracellular vesicles are key players in pathology of malaria and this includes *P. vivax* where concentration of circulating microparticles were associated with acute infections (Campos et al., 2010 Mal J). Moreover, Dr. Marti has pioneered this field since the original manuscript describing the role of EVs in malaria as intercellular communicators (Mantel et al., 2013 Cell). More recently, his group also demonstrated that interaction of EVs with bone marrow endothelial cells induce expression of IL-6 and IL-1 as well as vascular endothelium perturbations after trans-endothelial electrical resistance experiments (Mantel et al., 2016 Nat Comm). Furthermore, another recent report showed the physiological role of EVs in vivax malaria by demonstrating that EV uptake by human spleen fibroblast induced nuclear translocation of the NF-κB transcriptional factor, concomitant with surface expression of ICAM-1, thus facilitating cytoadherence of infected reticulocytes from *P. vivax* patients (Toda et al., 2020 Nat Comm). This growing evidence indicates that plasma circulating EVs are key communicators in malaria infections potentially explaining some of the findings reported in this work. Neglecting the importance of EVs in the discussion of this article is not reasonable and weakens this manuscript. Including a paragraph on EVs and accurate references in the discussion is thus strongly recommended.

We agree with the reviewer that extracellular vesicles are key communicators in malaria infection. We have not measured them in our study, however, and therefore can only speculate about their impact on our observations. We have added a phrase in the discussion:

“It is likely that other circulating factors that we have not directly measured in our study are also contributing to EC activation and vascular permeability. In particular, extracellular vesicles (EV) originating from ECs, platelets, and RBCs are present during malaria infection and are known to modulate the host immune response to the parasite^54-56^. In *P. falciparum*, infected RBCs release EVs containing immunogenic parasite antigens, that activate macrophages, induce neutrophil migration and alter endothelial barrier function^54,55^. In *P. vivax*, plasma-derived EVs from iRBCs are taken up by human spleen ﬁbroblasts (hSFs). This event signals NF-κB translocation and upregulation of ICAM-1 expression, facilitating cytoadherence of *P. vivax*-infected reticulocytes^56^.”

Minor comments:

1. The lack of a group including severe vivax malaria patients is a drawback of this article as this group would have firmly validated the predictor of severe disease.

This study was investigating a cohort of uncomplicated *P. vivax* malaria compared to controls. We agree that it will be important to extent our analysis to severe vivax malaria in future studies.

2. In the selection criteria of the patients to be included in the study, no information on other co-infections were mentioned. Is this information available? If so, this should be mentioned.

As described in the Methods sections, Page 6, line 132, mono infection by *P. vivax* was confirmed by analysis of blood smears and quantitative PCR (qPCR) for both *P. vivax* and *P. falciparum*. We agree that excluding other coinfections could have been of interest. However, the differential diagnosis for an acute febrile illness is very broad and it would be impractical to track all other possible diseases. In addition, the patients included in the present work had mild disease, and therefore were discharged from hospital after a positive malaria diagnosis. No further investigation on other infections was done.

The main coinfection to be considered for an acute febrile illness with no localizing signs in our context is Dengue Fever. Although Dengue coinfection in our cohort is possible, the incidence at the Hospital is only 2.8% (*P. vivax*/Dengue coinfection) (Magalhães et al., Plos NTD 2014). Thus, it is unlikely that such a coinfection would have a major impact on our findings.

3. This work determined the levels of PvLDH in a cohort of uncomplicated P. vivax patients as well as healthy volunteers using a double-sandwich ELISA assay: (i) are the clones to determine PvLDH values freely available to facilitate similar studies by independent groups? (ii) How was the cut-off of positivity defined? This is not evident, neither in the Materials and methods, nor in the results.

Clones are commercially available and were purchased from Vista Diagnostics International LLC, WA, USA. Information has been amended to the text in the Methods section.

*“*Cut-off of positivity was defined by correcting absorbance values generated in the plasma samples from healthy donors (controls) by blank values (plate controls), with both values being in the same range. Absorbance values higher than controls were considered positive. In parallel, we used schizont extracts to perform standard curves and lower absorbance values were in the range of O.D = 0.03-0.04. All positive patient samples gave O.D. values equal or higher than 0.05.”

4. It is not clear why varying percentages of pooled plasma (30% for imaging and flow cytometry, and 20% for impedance changes) from the different clusters were used for the functional EC assays. Moreover, no information about the concentration of plasma used for transcriptional analysis is available. Please clarify.

The concentration of 30% pooled plasma was also used for transcriptional analysis, as indicated in the Methods section, page 11, line 250. This information was also added in the legend of Figure 5B. We had run several optimisation time-course and titration experiments with individual plasma samples, testing concentrations of plasma varying from 10% up to 30% v/v and we did not observe differences in mRNA expression between 20% and 30% v/v plasma conditions.

As for the ECIS, our collaborators (Erich V de Paula group) have optimised this assay and they use a range of 15 to 20% (Santaterra et al. 2020). Higher concentrations of plasma reduces the reproducibility, probably to fibrin formation.

5. Reference 9 is a nonhuman primate study where no LDH is used. Please remove it.

Reference 9 has been removed following the reviewer suggestion.

6. Reference 39 is a review on the subject and cannot be included in the sentence on line 556 In agreement with a previous study8,39, where reference 8 is accurate. Please remove reference 39 from here.

The text has been amended as suggested.

Reviewer #3 (Significance (Required)):This paper further contributes to explain the conundrum of low peripheral blood parasitemia and clinical severity in *P. vivax*. Moreover, by including new human markers and solidly applying computational tools, this paper further contributes to advance clinical research in *P. vivax*.Clinical diagnosis of hematological disorders including anemia, lymphopenia and thrombocytopenia, are routinely obtained from a complete blood count. Therefore, I believe the major significance of this work is to raise public health awareness of including in these clinical examinations, the determination of PvLDH levels. They might prognose, as suggested by the authors, better diagnosis and treatment of *P. vivax*,My main expertise is the biology of host-pathogen interactions with a focus on *P. vivax*.Reviewer #4 (Evidence, reproducibility and clarity (Required)):The study evaluates *P. vivax* biomass (serum LDH) versus peripheral parasitemia with multiple variables. From the biomass Vivax high vs. Vivax low, they compare multiple determination in patients with uncomplicated *P. vivax*. This raises questions about disease and the presence of parasites in various organs. The question is if *P. vivax* sequesters and the answer is yes in the bone marrow and spleen. Does it sequester like *P. falciparum* that causes disease by sequestration by binding endothelium in various organs. That is less clear. As *P. vivax* is rarely fatal, the sequestration has not been studied. The presence of parasites in organs of *P. vivax* infected splenectomized squirrel and Aotus monkeys has been found in bone marrow and liver (note: splenecotomized monkeys so parasitemia can rise to higher levels than in non-splenectomized monkeys). There are studies of binding of schizonts infected red cells to lung endothelium in vitro does not answer the question of whether sequestration occurs in vivo.The most important complication of *P. vivax* is generally anemia. This did not correlate with vivax biomass, but this raises the question of the length of infection and the possibility that parasite biomass may vary at different times of infection. Anemia was seen in *P. vivax* infected patients, but it did not relate to biomass at the time of study. Note the caveat mentioned in the previous sentence on long term effects of infection on anemia.The finding of biomass with reduced platelet counts and endothelial effects that may be related to a serum factor and not sequestration. This is the main limitation of the paper besides the unknown long term effect infection. If one could identify an effect of *P. vivax* infected human serum, this may be worth a study in the future on what is in serum causing the effects.Reviewer #4 (Significance (Required)):This study is unique with the caveats mentioned above. It has a good review of the literature.

We appreciate the reviewer comments. In our cohort, the frequency of anaemia was not as high or severe as the frequency of thrombocytopenia and lymphopenia. However, we still find associations between endothelial cell activation marker Ang-2 and the pro-inflammatory cytokine IL-1 IL-1 negatively associated with several markers of anaemia, such as haemoglobin, haematocrit and RBC numbers. Although we did not further investigate this association, it may indicate indirect effects of parasite biomass on anaemia mediated by inflammation and EC activation, which will be further investigated in other current longitudinal cohort studies.

[Editors' note: further revisions were suggested prior to acceptance, as described below.]

Reviewer #4:The manuscript by by Silva- Filho et al., entitled: "Total parasite biomass but not peripheral parasitaemia is associated with endothelial and haematological perturbations in Plasmodium vivax patients" reports the importance of total parasite mass of P. vivax infection, and its relationship with a variety of host hematological and immunological markers. The study classifies infection based on clinical features, parasitemia, and host features. I have the following questions:1. The use of PvLDH in patient plasma as a proxy for parasite biomass has been established before. But the use of the biomass marker in combination with other host markers are novel. Can the authors explains more in-depth why the suspected deep tissue mass contribute to the PvLDH levels from different host tissues? Because the conclusion of this study relies on this parameter, an in-depth understanding of this marker is necessary.

Low peripheral parasitaemia of *P. vivax* infections (generally below 2%) compared to *P. falciparum* has been attributed to the strict tropism of *P. vivax* to infect mostly immature reticulocytes, a red blood cell (RBC) population that it is largely conﬁned to the erythropoietic niche of the bone marrow (~0.016% of all immature reticulocytes are in the circulation) [1, 2]. Such low peripheral parasitaemia has long been considered to indicate a low total parasite biomass in vivax malaria. More recently, existence of a significant *P. vivax* reservoir outside of circulation was predicted by disproportionately high PvLDH levels in peripheral circulation compared to parasitemia by blood smear (in particular in patients with complicated outcomes), and by modelling using experimental *P. cynomolgy* infections in non-human primates [3, 4]. A series of studies, including from our labs, have meanwhile provided direct evidence that bone marrow and spleen represent the major reservoir of parasite biomass in *P. vivax* infection [5-9].

PvLDH is produced by viable or recently killed parasites and hence considered a proxy for ongoing rather than past infection [3, 10]. PvLDH antigen capture ELISA established a direct relationship between pLDH levels and *P. vivax* parasitemia in ex vivo experiments, demonstrating that pLDH reflects total *P. vivax* parasite biomass [10]. Our study further explores the relevance of PvLDH beyond determination of parasite biomass, establishing it as a prognostic marker of host perturbations and disease severity, with a particular focus on markers of changes in the hematopoietic niches of bone marrow and spleen.

We have amended the text in the Introduction and Discussion sections as indicated below:

“A series of recent studies in acute and chronic human *P. vivax* infection have meanwhile provided direct evidence that bone marrow and spleen represent the major reservoir of parasite biomass in *P. vivax* infection ^17,20-22^.”

“The existence of a significant *P. vivax* reservoir outside of circulation was first predicted by disproportionately high PvLDH levels in peripheral circulation compared to parasitemia by blood smear (in particular in patients with complicated outcomes), and by modelling using experimental *P. cynomolgy* infections in non-human primates^8,36^. Recent studies provide direct evidence that bone marrow and spleen represent the major reservoir of parasite biomass in *P. vivax* infection^11,17,20-22^. PvLDH is produced by viable or recently killed parasites and hence considered a proxy for ongoing rather than past infection^8,48^. PvLDH antigen capture ELISA established a direct relationship between pLDH levels and *P. vivax* parasitemia in ex vivo experiments, demonstrating that pLDH reflects total *P. vivax* parasite biomass^48^. Our study further explores the relevance of PvLDH as a prognostic marker of host perturbations and disease severity, with a particular focus on markers of changes in the hematopoietic niches of bone marrow and spleen.”

2. One of the important conclusion is the severity of the infection is parasite-biomass dependent. While the current study does not include severe vivax cases, are there studies support the author's conclusion?

This is a key question. In contrast to *P. falciparum*-infected individuals, a wide range of complicated clinical syndromes occurs in *P. vivax* patients even at low or subpatent parasitemia [11] – thus indicating that peripheral parasitemia is a poor predictor of clinical outcomes. Two lines of evidence support our conclusion that severity of infection is dependent on parasite biomass instead. First, the discrepancy between PvLDH levels and peripheral parasitaemia determined by blood smears is more evident in *P. vivax*-infected patients with complicated outcomes: the ratio of plasma pLDH to peripheral parasitaemia is 6-fold higher than in non-severe patients. The same comparison between severe and non-severe *P. falciparum* patients reveals only a 1.4-fold difference [3]. Second, severe thrombocytopenia (platelet counts under 50,000/μL) and lymphopenia (lymphocyte counts under 500/μL) have been associated with severity in vivax patients, suggesting that these haematological complications could be explored as a marker of severity for this species [12-15]. Both thrombocytopenia and lymphopenia were more frequent in patients in the Cluster 2 (Vivax^high^) in our study. By integrating these clinical perturbations with host biomarker measurements and parasite parameters, we demonstrated the high attribute value of total parasite biomass in predicting the severity of thrombocytopenia and lymphopenia (Figures S7 and S8), and highly significant correlations with endothelial activation, glycocalyx breakdown and other markers of inflammation (Figure 6). Our ongoing studies aim to directly investigate the role of host parasite interactions in the hematopoietic niches in *P. vivax* pathogenesis and severity.

The following text has been added in the Discussion section:

“In contrast to *P. falciparum*-infected individuals, a wide range of complicated clinical syndromes occurs in *P. vivax* patients even at low or subpatent parasitemia^53^ – thus indicating that peripheral parasitemia is a poor predictor of clinical outcomes. Two lines of evidence support our conclusion that severity of infection is dependent on parasite biomass instead. First, the discrepancy between PvLDH levels and peripheral parasitaemia determined by blood smears is more evident in *P. vivax*-infected patients with complicated outcomes: the ratio of plasma pLDH to peripheral parasitaemia is 6-fold higher than in non-severe patients. The same comparison between severe and non-severe *P. falciparum* patients reveals only a 1.4-fold difference^8^. Second, although thrombocytopenia and lymphopenia are not included in the World Health Organization (WHO) criteria for defining severe malaria, it has been associated with severe manifestations and the need for blood and platelet transfusions in severe vivax malaria. This points out their clinical relevance in malaria diagnosis and treatment ^24,25,54-56^, suggesting that these haematological complications could be explored as markers of severity for this species. Both severe thrombocytopenia and lymphopenia were more frequent in patients in the Cluster 2 (Vivax^high^) in our study. By integrating these clinical perturbations with host biomarker measurements and parasite parameters, we demonstrated the high attribute value of total parasite biomass in predicting the severity of thrombocytopenia and lymphopenia and highly significant correlations with endothelial activation, glycocalyx breakdown and other markers of inflammation.”

3. Results Page 17. The authors found that the total parasite biomass was higher and not correlated with parasitemia, particularly in patients of cluster 2. Why is cluster 2 strong in this aspect?

The highly significant and positive association between endothelial activation, Syndecan-1 and parasite biomass (PvLDH) indicates a positive feedback loop between glycocalyx breakdown, activation of endothelial receptors such as ICAM-1and VCAM-1 and parasite accumulation in deep tissues. Patients in Cluster 2 show higher plasma levels of all these markers. Consistent with previous reports [16, 17], we propose that elevated EC activation and glycocalyx damage increases the exposure of adhesion molecules, which in turn favours endothelial cytoadherence of *P. vivax*-infected RBCs, in particular in the splenic red pulp cords and in the BM [18-20]. Accordingly, application of a best-fit classification tree model identifies Syndecan-1 is a putative host biomarker (EC glycocalyx breakdown marker) predicting total parasite biomass in *P. vivax* patients (Figure 6D). We hypothesise that elevated endothelial activation and damage in patients of Cluster 2 results in increased cytoadherence of *P. vivax* iRBCs and hence accumulation and growth in deep tissues – thus reducing the fraction of the parasite biomass in circulation.

The following text has been added in the Discussion section:

“Furthermore, the highly significant and positive associations between endothelial activation, Syndecan-1 and parasite biomass (PvLDH) indicates a positive feedback loop between glycocalyx breakdown, activation of endothelial receptors such as ICAM-1and VCAM-1 and parasite accumulation in deep tissues. Vivax^High^ patients show higher plasma levels of all these markers. Consistent with previous reports ^44,49^, we propose that elevated EC activation and glycocalyx damage increases the exposure of adhesion molecules, which in turn favours endothelial cytoadherence of *P. vivax*-infected RBCs, in particular in the splenic red pulp cords and in the BM ^21,50-52^. Accordingly, application of a best-fit classification tree model identifies Syndecan-1 is a putative host biomarker (EC glycocalyx breakdown marker) predicting total parasite biomass in *P. vivax* patients. We hypothesise that elevated endothelial activation and damage in Vivax^High^ patients results in increased cytoadherence of *P. vivax* iRBCs and hence accumulation and growth in deep tissues, thus reducing the fraction of the parasite biomass in circulation.”

4. Network and module analysis. The results seem to show that the healthy donor and the patients have different network structure and module numbers? Can the authors explain more on it?

We have included in the Methods and Results sections details about the network topology analysis using the NetworkAnalyzer, jActiveModules and MCODE plugins in Cytoscape. We have also added the supplementary Table S4 containing all parameters for the comparative network topology analysis between the graphs for healthy donors and *P. vivax* patients. Detailed descriptions of all analyzed parameters can be found in the references 33 and 34 in the main manuscript text.

The observed difference in the network structure is largely due to the introduction of parasite parameters in the patient graph. A major observation in the network graph of *P. vivax* patients is the central position of the total parasite biomass marker PvLDH, due to its equally strong interactions with the two main functional modules 1 and 2 (Figure 6A). Given that the hematopoietic niches of the BM and the spleen are the major reservoir of total parasite biomass, interactions of PvLDH with these two main modules indicate an interplay between parasite infection in these niches and endothelial activation/damage as well as the proinflammatory response that results in myeloid-biased differentiation, thrombocytopenia and lymphopenia.

Similar to a previous study with *P. vivax* patients and healthy donors from an endemic area in Brazil [21], our analysis revealed a dense network of interactions with homogenous and centralized topology among the biomarkers in healthy donors (see values for network diameter; heterogeneity; density and connectivity in Table S4). The network topology is drastically altered in symptomatic *P. vivax* patients, with a decentralized topology and lower complexity including reduced number of significant interactions between parameters (91 pairwise connections in *P. vivax* patients vs 166 pairwise connections in healthy donors; p<0.0001) (Figure 6A, Table S4). Interestingly, due to its decentralized and heterogenous pattern of interactions, the network graph of *P. vivax* patients results in clear functional modules and more closely connected biomarkers. Of note, the network pattern described in our study is similar to protein-protein associated networks described previously in *P. vivax* malaria and in other clinical contexts [21, 22].

The following text has been added in the Methods, Results and Discussion sections:

“Network topology and module analysis were performed using the NetworkAnalyzer, jActiveModules and MCODE plugins in Cytoscape ^72,73^. Supplementary File1 shows the results for all parameters quantified in the comparative network topology analysis between the graphs for healthy donors and *P. vivax* patients.

Page 12, line 285: Similar to a previous study with *P. vivax* patients and healthy donors from an endemic area in Brazil ^46^, our analysis revealed a dense network of interactions with homogenous and centralized topology among the biomarkers in healthy donors (Figure 6A, Supplementary File 1). The network topology is drastically altered in symptomatic *P. vivax* patients, largely due to the introduction of parasite parameters in the patient graph (Figure 6A, Supplementary File 1).”

“Of note, the network pattern described in our study is similar to protein-protein associated networks described previously in *P. vivax* malaria and in other clinical contexts^46,47^.”

“A major observation in the network graph of *P. vivax* patients is the central position of the total parasite biomass marker PvLDH, due to its equally strong interactions with the two main functional modules 1 and 2. Given that the hematopoietic niches of the BM and the spleen are the major reservoir of parasite biomass, interactions of PvLDH with these two main modules indicate an interplay between parasite infection in these niches and endothelial activation/damage as well as the proinflammatory response that results in myeloid-biased differentiation, thrombocytopenia and lymphopenia.”

References

1. Kanjee, U., et al., Molecular and cellular interactions defining the tropism of Plasmodium vivax for reticulocytes. Curr Opin Microbiol, 2018. 46: p. 109-115.

2. Ovchynnikova, E., et al., DARC extracellular domain remodeling in maturating reticulocytes explains. Blood, 2017. 130(12): p. 1441-1444.

3. Barber, B.E., et al., Parasite biomass-related inflammation, endothelial activation, microvascular dysfunction and disease severity in vivax malaria. PLoS Pathog, 2015. 11(1): p. e1004558.

4. Fonseca, L.L., et al., A model of Plasmodium vivax concealment based on Plasmodium cynomolgi infections in Macaca mulatta. Malar J, 2017. 16(1): p. 375.

5. Obaldia, N., et al., Bone Marrow Is a Major Parasite Reservoir in Plasmodium vivax Infection. MBio, 2018. 9(3).

6. Baro, B., et al., Plasmodium vivax gametocytes in the bone marrow of an acute malaria patient and changes in the erythroid miRNA profile. PLoS Negl Trop Dis, 2017. 11(4): p. e0005365.

7. Brito, M.A.M., et al., Morphological and Transcriptional Changes in Human Bone Marrow During Natural Plasmodium vivax Malaria Infections. J Infect Dis, 2020.

8. Kho, S., et al., Evaluation of splenic accumulation and colocalization of immature reticulocytes and Plasmodium vivax in asymptomatic malaria: A prospective human splenectomy study. PLoS Med, 2021. 18(5): p. e1003632.

9. Kho, S., et al., Hidden Biomass of Intact Malaria Parasites in the Human Spleen. N Engl J Med, 2021. 384(21): p. 2067-2069.

10. Druilhe, P., et al., Improved assessment of plasmodium vivax response to antimalarial drugs by a colorimetric double-site plasmodium lactate dehydrogenase antigen capture enzyme-linked immunosorbent assay. Antimicrob Agents Chemother, 2007. 51(6): p. 2112-6.

11. Baird, J.K., Evidence and implications of mortality associated with acute Plasmodium vivax malaria. Clin Microbiol Rev, 2013. 26(1): p. 36-57.

12. Gerardin, P., et al., Prognostic value of thrombocytopenia in African children with falciparum malaria. Am J Trop Med Hyg, 2002. 66(6): p. 686-91.

13. Kochar, D.K., et al., Thrombocytopenia in Plasmodium falciparum, Plasmodium vivax and mixed infection malaria: a study from Bikaner (Northwestern India). Platelets, 2010. 21(8): p. 623-7.

14. Kochar, D.K., et al., Plasmodium vivax malaria. Emerg Infect Dis, 2005. 11(1): p. 132-4.

15. Andrade, B.B., et al., Severe Plasmodium vivax malaria exhibits marked inflammatory imbalance. Malar J, 2010. 9: p. 13.

16. Yeo, T.W., et al., Glycocalyx Breakdown Is Associated With Severe Disease and Fatal Outcome in Plasmodium falciparum Malaria. Clin Infect Dis, 2019. 69(10): p. 1712-1720.

17. Barber, B.E., et al., Endothelial glycocalyx degradation and disease severity in Plasmodium vivax and Plasmodium knowlesi malaria. Sci Rep, 2021. 11(1): p. 9741.

18. Introini, V., et al., Endothelial glycocalyx regulates cytoadherence in Plasmodium falciparum malaria. J R Soc Interface, 2018. 15(149): p. 20180773.

19. Hempel, C., et al., Binding of Plasmodium falciparum to CD36 can be shielded by the glycocalyx. Malar J, 2017. 16(1): p. 193.

20. Toda, H., et al., Plasma-derived extracellular vesicles from Plasmodium vivax patients signal spleen fibroblasts via NF-κB facilitating parasite cytoadherence. Nat Commun, 2020. 11(1): p. 2761.

21. Mendonça, V.R., et al., Networking the host immune response in Plasmodium vivax malaria. Malar J, 2013. 12: p. 69.

22. Frankenstein, Z., U. Alon, and I.R. Cohen, The immune-body cytokine network defines a social architecture of cell interactions. Biol Direct, 2006. 1: p. 32.